# Self-Supervised Learning from Structural Invariance

**Yipeng Zhang**[1,2†]**, Hafez Ghaemi**[1,2]**, Jungyoon Lee**[1,2] **& Laurent Charlin**[1,2,3,4]
[1]Mila - Québec AI Institute, [2]Université de Montréal, [3]HEC Montréal, [4]CIFAR AI Chair
Montréal, Canada

## Abstract

Joint-embedding *self-supervised learning* (SSL), the key paradigm for unsupervised representation learning from visual data, learns from invariances between semantically-related data pairs. We study the one-to-many mapping problem in SSL, where each datum may be mapped to multiple valid targets. This arises when data pairs come from naturally occurring generative processes, e.g., successive video frames. We show that existing methods struggle to flexibly capture this conditional uncertainty. As a remedy, we introduce a variational distribution that models this uncertainty in the latent space, and derive a lower bound on the pairwise mutual information. We also propose a simpler variant of the same idea using sparsity regularization. Our model, AdaSSL, applies to both contrastive and predictive SSL methods, and we empirically show its advantages on identifiability, generalization, fine-grained image understanding, and world modeling on videos.[2]

## 1 Introduction

Over the last decade, joint-embedding *self-supervised learning* (SSL) has become the dominant approach in representation learning from unlabeled visual data (Chen et al., 2020a; Zbontar et al., 2021; Grill et al., 2020; Radford et al., 2021; Assran et al., 2023). The intuition behind SSL is to obtain semantically-related data pairs, often called *positive pairs*, and encourage their representations to be similar, with proper regularization to prevent the encoder collapsing to a constant function (Wang & Isola, 2020; Garrido et al., 2023a; Zhuo et al., 2023).

Positive pairs are typically built with handcrafted augmentations (e.g., cropping, color jittering), which perturb pixels while preserving semantics. Such augmentations cannot precisely mimic changes in natural factors of variation that drive real-world distribution shifts (Ibrahim et al., 2023). For instance, rotating an image moves the entire scene rather than a single object. Consequently, augmentations may fail to induce the right invariances (Ibrahim et al., 2023, 2022; Bouchacourt et al., 2021), discard fine-grained information (Chen et al., 2020a; Zhang et al., 2024), and require modality-specific heuristics (Balestriero et al., 2023) and incur additional computation burden (Bordes et al., 2023), ultimately harming downstream performance.

One alternative is to exploit naturally-paired data—nearby video frames (Klindt et al., 2021; Bardes et al., 2024; Sermanet et al., 2018), image–caption pairs (Radford et al., 2021), class labels (Khosla et al., 2020), or embeddings from other models (Sobal et al., 2025; Feizi et al., 2024)—which better reflect real-world variations. From the lens of *causal representation learning* (CRL) (Yao et al., 2025; Reizinger et al., 2025), positive pairs $(\mathbf{x}, \mathbf{x}^+)$ are deterministically mapped from latent factors sampled according to $(\mathbf{z}, \mathbf{z}^+) \sim p(\mathbf{z})p(\mathbf{z}^+ \mid \mathbf{z})$. Unlike augmentations that operate in observation space, natural positive pairs differ according to structured changes in latent factors of the *data generating process* (DGP). Modelling these latent changes often improves generalization (Ibrahim et al.,

---

[†]Correspondence to `yipeng.zhang@mila.quebec`.
[2]Code and the most recent version of this paper can be found at `https://github.com/SkrighYZ/AdaSSL`.

Preprint.

2022; Dittadi et al., 2021; Kaur et al., 2023) and visual understanding (Awal et al., 2024; Garrido et al., 2025; Lippe et al., 2023).

Despite benefits, leveraging natural pairs for SSL remains challenging because they also induce complex conditional distributions $p(\mathbf{z}^+ \mid \mathbf{z})$. In world modeling (Ha & Schmidhuber, 2018b,a; Hafner et al., 2025; Assran et al., 2025), the present state may lead to multiple plausible futures (e.g., a car may turn left or right), making the conditional distribution inherently multimodal. For image–caption pairs, caption details vary with image complexity, producing heteroscedastic noise. SSL methods that fail to capture this uncertainty often discard information not shared between the pair, leading to degraded performance (Chen et al., 2020a; Radford et al., 2021; Jing et al., 2022; Yuksekgonul et al., 2023; Trusca et al., 2024; Zhang et al., 2024). We argue that leveraging the structure of $p(\mathbf{z}^+ \mid \mathbf{z})$ enables SSL to learn more generalizable features—a principle we call **SSL from structural invariance**.

Building on recent advances that enable SSL models to learn $p(\mathbf{z}^+ \mid \mathbf{z})$ that has constant, anisotropic noise (Kügelgen et al., 2021; Zimmermann et al., 2021; Rusak et al., 2025), we provide a solution to model unknown, potentially complex conditional distributions in SSL. We take inspiration from *joint-embedding predictive architectures* (JEPAs) (LeCun, 2022; Garrido et al., 2024; Assran et al., 2025), which use a latent variable that captures the uncertainty in predictions. In contrast to prior work (Devillers & Lefort, 2023; Garrido et al., 2024; Ghaemi et al., 2024; Dangovski et al., 2022), we do not assume access to this variable and infer it purely from the structure hidden in positive pairs. For contrastive learning, we derive a tractable lower bound on the mutual information between the paired views, and we empirically show our modification is compatible with non-contrastive methods. We name our method **Adaptive SSL (AdaSSL)** as it adapts to different conditional distributions.

We evaluate AdaSSL in controlled settings with numerical data, natural images, and videos. On numerical data, we show that existing SSL methods lack the ability to model non-trivial conditionals, and AdaSSL achieves better performance both in- and out-of-distribution (OOD). On images, AdaSSL consistently recovers fine-grained features better than baselines. On videos, AdaSSL captures stochastic object accelerations that baselines discard without sacrificing class accuracy.

## 2  Method

In this section, we describe the proposed method. We present preliminaries of SSL in §A and derivations and technical details of our method in §B.

### 2.1  Data generating process

In CRL, representation learning is viewed as learning to *invert* the true DGP (Reizinger et al., 2025; Zimmermann et al., 2021). In SSL, we assume a data pair $\mathbf{x}, \mathbf{x}^+$ follows this generative process:

$$\mathbf{z} \sim p(\mathbf{z}), \quad \mathbf{z}^+ \mid \mathbf{z} \sim p(\mathbf{z}^+ \mid \mathbf{z}), \quad \mathbf{x} = g(\mathbf{z}), \quad \mathbf{x}^+ = g(\mathbf{z}^+), \tag{1}$$

where $g : \mathcal{Z} \to \mathcal{X}$ is an unknown mixing function that produces the observations $\mathbf{x}, \mathbf{x}^+ \in \mathcal{X}$ based on the latent factors $\mathbf{z}, \mathbf{z}^+ \in \mathbb{R}^{d_z}$. The goal is to learn a function $f : \mathcal{X} \to \mathbb{R}^{d_f}$ that encodes the data into an embedding space $\mathbb{R}^{d_f}$ such that we can predict a subset[3] of the latent factors that are useful for downstream tasks from $f(\mathbf{x})$ with a simple function, e.g., an affine transformation. We denote this subset of latent factors as "content factors" $\mathbf{c} := \mathbf{z}_{\mathbb{I}}$ for $\mathbb{I} \subseteq [d_z]$, and the other (less relevant) factors as "style" factors $\mathbf{s} := \mathbf{z}_{[d_z] \setminus \mathbb{I}}$ following Kügelgen et al. (2021).

### 2.2  Modeling complex conditionals with a latent variable

To capture the complex conditional distributions $p(\mathbf{x}^+ \mid \mathbf{x})$, a pushforward of $p(\mathbf{z}^+ \mid \mathbf{z})$ through $g$, we use a latent variable $\mathbf{r}$ to model information about $\mathbf{x}^+$ that cannot be solely predicted from $\mathbf{x}$. Learning a representation that maximally preserves the mutual information (MI) between paired embeddings is useful for representation learning (Linsker, 1988; Tschannen et al., 2020; Oord et al., 2018). It also provides a way to interpret the desiderata of $\mathbf{r}$. Specifically, by the chain rule of MI,

$$I(\mathbf{x}; \mathbf{x}^+) = I(\mathbf{x}, \mathbf{r}; \mathbf{x}^+) - I(\mathbf{r}; \mathbf{x}^+ \mid \mathbf{x}). \tag{2}$$

---

[3]Although full latent recovery is often the goal in theory, invariance to certain style factors in practice can help generalization (Deng et al., 2022) and prevent shortcut solutions in SSL (Chen et al., 2020a).

Intuitively, $\mathbf{r}$ should help $\mathbf{x}$ predict $\mathbf{x}^+$ without simply copying $\mathbf{x}^+$. This motivates the general form of our objective:

$$\mathcal{L}_{\mathrm{AdaSSL}} = \mathcal{L}_{\mathrm{SSL}}((\mathbf{x}, \mathbf{r}), \mathbf{x}^+) + \beta \mathcal{L}_{\mathrm{Reg}}(\mathbf{r}), \tag{3}$$

where the SSL term is any standard SSL loss (e.g., $\mathcal{L}_{\mathrm{InfoNCE}}$) that encourages $\mathbf{r}$ to aid prediction of $\mathbf{x}^+$ while the regularizer penalizes $\mathbf{r}$ from becoming an unrestricted shortcut. The hyperparameter $\beta$ controls the strength of regularization per standard practice (Higgins et al., 2017; Locatello et al., 2020). This objective matches the conceptual framework depicted in Fig. 13 of LeCun (2022).

## 2.3 AdaSSL

**AdaSSL-V and a lower bound on $I(\mathbf{x}, \mathbf{x}^+)$.** We first learn the posterior $p(\mathbf{r} \mid \mathbf{x}, \mathbf{x}^+)$ with a variational distribution $q_\phi(\mathbf{r} \mid \mathbf{x}, \mathbf{x}^+)$ (Kingma & Welling, 2014; Sohn et al., 2015). The joint then becomes $\tilde{p}(\mathbf{x}, \mathbf{x}^+, \mathbf{r}) := p(\mathbf{x}, \mathbf{x}^+) q(\mathbf{r} \mid \mathbf{x}, \mathbf{x}^+)$. The informational-theoretical properties of contrastive learning allow us to optimize a lower bound on $I(\mathbf{x}, \mathbf{x}^+)$[4]:

$$\mathcal{L}_{\mathrm{AdaSSL-V}} = \mathcal{L}_{\mathrm{SSL}}\big(\mathbb{E}_{q_\phi} \psi_1(\mathbf{x}, \mathbf{r}), \psi_2(\mathbf{x}^+)\big) + \beta D_{\mathrm{KL}}(q_\phi(\mathbf{r} \mid \mathbf{x}, \mathbf{x}^+) \| p_\theta(\mathbf{r} \mid \mathbf{x})). \tag{4}$$

In practice, we parameterize $q_\phi$ and $p_\theta$ using lightweight MLPs on top of the embeddings $f(\mathbf{x})$ and $f(\mathbf{x}^+)$, modeling both as factorized Gaussians. $\psi_1(\mathbf{x}, \mathbf{r})$ uses $\mathbf{r} \sim q_\phi$ to *edit* the embedding $f(\mathbf{x})$ with a linear or MLP editor $t$, and $\psi_2(\mathbf{x}^+) = \frac{f(\mathbf{x}^+)}{\|f(\mathbf{x}^+)\|_2}$. We call this method **AdaSSL-V**(variational).

**AdaSSL-S and sparse modular edits.** Natural transitions usually correspond to sparse changes in the latent factors, an inductive bias widely adopted in the identifiability literature (Ahuja et al., 2022; Klindt et al., 2021; Lippe et al., 2023). Therefore, we hypothesize that we can implement Eq. 3 by predicting $\mathbf{r}$ and regularizing its sparsity. **AdaSSL-S**(parse) realizes this idea. Instead of learning a variational posterior, we predict $\mathbf{r}$ deterministically from $f(\mathbf{x})$ and $f(\mathbf{x}^+)$: $\mathbf{r} = m(f(\mathbf{x}), f(\mathbf{x}^+))$, where $m$ is an MLP followed by `tanh` activation. We then regularize the sparsity of $\mathbf{r}$:

$$\mathcal{L}_{\mathrm{AdaSSL-S}} = \mathcal{L}_{\mathrm{SSL}}\big(\psi_1(\mathbf{x}, \mathbf{r}), \psi_2(\mathbf{x}^+)\big) + \beta \|\mathbf{r}\|_0, \tag{5}$$

where the $L_0$ penalty is made differentiable through the Gumbel-Sigmoid estimator similar to the one used by Lachapelle et al. (2022); Brouillard et al. (2020). Inspired by Ibrahim et al. (2022); Hu et al. (2022), we use a modular editing function $t$ in $\psi_1$.

*Remark.* AdaSSL-V and AdaSSL-S are applicable to any SSL method because they address the limitation of the invariance part of their objectives. We refer readers to §B for further details.

## 3 Experiments

We evaluate AdaSSL on numerical data (§3.2), natural images (§3.3), and videos (§3.3) to test its ability to learn generalizable features. Additionally, in §C, we show that our method performs full latent recovery and disentanglement better than existing methods.

### 3.1 Overview of experimental protocol

**Baselines.** Our experiments in §3.2 and §3.3 focus on contrastive SSL. InfoNCE (Chen et al., 2020a; Oord et al., 2018) and AnInfoNCE (Rusak et al., 2025) are the contrastive baselines that account for isotropic and anisotropic noise in $p(\mathbf{z}^+ \mid \mathbf{z})$, respectively (details in §A.1). AnInfoNCE learns directional weights of the similarity function, $\mathbf{\Lambda}$. For a fair comparison, we also use a learnable scalar weight $\lambda$ for other methods in §3.2 and §3.3 and find it beneficial. Table 5 compares the similarity functions across methods. For the video experiments in §3.3, we use BYOL (Grill et al., 2020) as our base SSL method.

**H-InfoNCE.** In addition to existing baselines, we introduce H-InfoNCE, which extend AnInfoNCE to account for heteroscedastic noise by predicting $\mathbf{\Lambda}_\mathbf{x}$ from $f(\mathbf{x})$ with an affine function (H-InfoNCE$_{\mathrm{Affine}}$) or an MLP (H-InfoNCE$_{\mathrm{MLP}}$); it replaces $\mathbf{\Lambda}$ in AnInfoNCE's similarity function with this conditional $\mathbf{\Lambda}_\mathbf{x}$. Additionally, H-InfoNCE uses another MLP predictor to predict $f(\mathbf{x}^+)$ from $f(\mathbf{x})$, similar to predictive SSL, except for in Table 1, where we ensure $\mathbb{E}[\mathbf{z}^+ \mid \mathbf{z}] = \mathbf{z}$.

---

[4] One can equivalently replace $I(\mathbf{x}; \mathbf{x}^+)$ with $I(f(\mathbf{x}); f(\mathbf{x}^+))$, since our method operates on paired embeddings. For simplicity, we use the notation $I(\mathbf{x}; \mathbf{x}^+)$ throughout, but in practice our method aims to maximize $I(f(\mathbf{x}); f(\mathbf{x}^+)) \le I(\mathbf{x}; \mathbf{x}^+)$.

Table 1: Linear regression $R^2$ on unimodal $p(\mathbf{z}^+ \mid \mathbf{z})$. All experiments share the same $\Sigma$ and the mixing function $g$ for each trial. Although all models achieve good performance on the training set $p(\mathbf{z})$, a flexible model is crucial to achieving good OOD performance. Values below 0.7 are dimmed.

| $\mathrm{Var}(\mathbf{c}^+ \mid \mathbf{c})$ | Model | MODEL SPACE: UNBOUNDED | | | MODEL SPACE: HYPERSPHERE | | |
|---|---|---|---|---|---|---|---|
| | | $p(\mathbf{z})$ | $\mathcal{N}(0,5\cdot\mathbf{I})$ | $\mathcal{N}(0,5\cdot\mathbf{I})_{\mathrm{OOD}}$ | $p(\mathbf{z})$ | $\mathcal{N}(0,5\cdot\mathbf{I})$ | $\mathcal{N}(0,5\cdot\mathbf{I})_{\mathrm{OOD}}$ |
| - | Identity | $0.7410_{\pm 0.0943}$ | $0.5103_{\pm 0.0374}$ | $0.1243_{\pm 0.0883}$ | $0.7410_{\pm 0.0943}$ | $0.5103_{\pm 0.0374}$ | $0.1243_{\pm 0.0883}$ |
| 0 | InfoNCE | $0.9912_{\pm 0.0051}$ | $0.9614_{\pm 0.0060}$ | $0.8924_{\pm 0.0590}$ | $0.8657_{\pm 0.1462}$ | $0.8004_{\pm 0.0764}$ | $0.2683_{\pm 0.2626}$ |
| 1 | InfoNCE | $0.9943_{\pm 0.0031}$ | $0.9731_{\pm 0.0070}$ | $0.9564_{\pm 0.0074}$ | $0.9785_{\pm 0.0178}$ | $0.9104_{\pm 0.0154}$ | $0.6944_{\pm 0.0657}$ |
| | H-InfoNCE$_{\mathrm{Affine}}$ | $0.9956_{\pm 0.0019}$ | $0.9736_{\pm 0.0080}$ | $0.9592_{\pm 0.0072}$ | $0.9953_{\pm 0.0021}$ | $0.9645_{\pm 0.0065}$ | $0.9154_{\pm 0.0100}$ |
| Anisotropic | InfoNCE | $0.9968_{\pm 0.0013}$ | $0.9764_{\pm 0.0055}$ | $0.9668_{\pm 0.0056}$ | $0.9509_{\pm 0.0358}$ | $0.7755_{\pm 0.1385}$ | $0.3523_{\pm 0.2323}$ |
| | AnInfoNCE | $0.9962_{\pm 0.0019}$ | $0.9753_{\pm 0.0068}$ | $0.9627_{\pm 0.0088}$ | $0.9613_{\pm 0.0418}$ | $0.8403_{\pm 0.0299}$ | $0.4022_{\pm 0.2316}$ |
| | H-InfoNCE$_{\mathrm{Affine}}$ | $0.9963_{\pm 0.0019}$ | $0.9685_{\pm 0.0032}$ | $0.9510_{\pm 0.0023}$ | $0.9970_{\pm 0.0017}$ | $0.9537_{\pm 0.0149}$ | $0.9018_{\pm 0.0035}$ |
| Heteroscedastic (affine+activation) | InfoNCE | $0.8553_{\pm 0.0532}$ | $0.2664_{\pm 0.0984}$ | $-0.1891_{\pm 0.2545}$ | $0.7851_{\pm 0.0920}$ | $0.2690_{\pm 0.1024}$ | $0.0209_{\pm 0.1110}$ |
| | AnInfoNCE | $0.8447_{\pm 0.0611}$ | $0.2745_{\pm 0.1052}$ | $-0.2277_{\pm 0.3284}$ | $0.7563_{\pm 0.1276}$ | $0.2563_{\pm 0.1092}$ | $0.0070_{\pm 0.1230}$ |
| | H-InfoNCE$_{\mathrm{Affine}}$ | $0.9826_{\pm 0.0060}$ | $0.9482_{\pm 0.0165}$ | $0.8666_{\pm 0.0741}$ | $0.9426_{\pm 0.0222}$ | $0.6276_{\pm 0.1084}$ | $0.3106_{\pm 0.1218}$ |
| | H-InfoNCE$_{\mathrm{MLP}}$ | $0.9892_{\pm 0.0023}$ | $0.9610_{\pm 0.0098}$ | $0.9149_{\pm 0.0348}$ | $0.9856_{\pm 0.0075}$ | $0.9288_{\pm 0.0175}$ | $0.7633_{\pm 0.0576}$ |

**Experimental setup.** We use a five-layer MLP as $f$ for the numerical experiments in §3.2, a ResNet-18 encoder followed by a two-layer MLP projector for the image experiments in §3.3, and a five-layer 3D CNN followed by a three-layer MLP projector for videos. Unless otherwise noted, we train the model from scratch on the training set and perform model selection based on the performance of an online affine probe on the validation set. For evaluation in §3.2, we follow Zimmermann et al. (2021) by training an affine probe on top of the *embeddings* produced by the frozen $f$ on the training data. For evaluation in §3.3, we train an affine probe on both the embeddings and the output of the frozen encoder, which we refer to as *representations*. We then evaluate the probes' performance on the test set following standard practice (Chen et al., 2020a; Grill et al., 2020). Additional experimental details can be found in §G.

## 3.2 Numerical data

In this section, we study the effect of complexity of the conditional variance in $p(\mathbf{z}^+ \mid \mathbf{z})$. Specifically, we sample correlated latents $\mathbf{c} \sim \mathcal{N}(0, \Sigma)$ and sample $\mathbf{c}^+$ from different conditional distributions $p(\mathbf{c}^+ \mid \mathbf{c})$. Style latents are sampled independently: $\mathbf{s}, \mathbf{s}^+ \sim \mathcal{N}(0, \mathbf{I})$, yielding $\mathbf{z} = [\mathbf{c}, \mathbf{s}]$ and $\mathbf{z}^+ = [\mathbf{c}^+, \mathbf{s}^+]$. A random invertible MLP parameterizing $g$ (details in §G.2) maps these latents to observations $\mathbf{x}, \mathbf{x}^+$ via Eq. 1. We then train linear

Table 2: Linear regression $R^2$ on complex $p(\mathbf{c}^+ \mid \mathbf{c})$. All models normalize embeddings and AdaSSL outperforms baselines.

| Model | $p(\mathbf{z})$ | $\mathcal{N}(0,5\cdot\mathbf{I})$ | $\mathcal{N}(0,5\cdot\mathbf{I})_{\mathrm{OOD}}$ |
|---|---|---|---|
| InfoNCE | $0.5210_{\pm 0.1611}$ | $0.5024_{\pm 0.0850}$ | $0.0395_{\pm 0.3141}$ |
| AnInfoNCE | $0.5446_{\pm 0.1745}$ | $0.5578_{\pm 0.1271}$ | $0.1652_{\pm 0.2261}$ |
| H-InfoNCE$_{\mathrm{MLP}}$ | $\underline{0.8750}_{\pm 0.0658}$ | $0.7784_{\pm 0.0915}$ | $0.5471_{\pm 0.2480}$ |
| AdaSSL-V | $0.8609_{\pm 0.0740}$ | $\mathbf{0.8656}_{\pm 0.0195}$ | $\mathbf{0.6638}_{\pm 0.0956}$ |
| AdaSSL-S | $\mathbf{0.9187}_{\pm 0.0174}$ | $\underline{0.8472}_{\pm 0.0292}$ | $\underline{0.6325}_{\pm 0.0737}$ |

regressors to predict $\mathbf{c}$ from $f(\mathbf{x}) = f(g([\mathbf{c}, \mathbf{s}]))$, where $f$ is the frozen encoder trained on $p(\mathbf{z})$. We perform three types of evaluation: (a) train and evaluate the regressor on $p(\mathbf{z})$, (b) train and evaluate the regressor on $\mathcal{N}(0, 5 \cdot \mathbf{I})$, and (c) train on $p(\mathbf{z})$ and test on $\mathcal{N}(0, 5 \cdot \mathbf{I})$ (denoted by $\mathcal{N}(0, 5 \cdot \mathbf{I})_{\mathrm{OOD}}$ in Table 1), where the latter two evaluate the representations' robustness under distribution shifts. Following prior works (Zimmermann et al., 2021; Kügelgen et al., 2021), we vary the latent space assumptions (unbounded or hypersphere) and model flexibility (InfoNCE, AnInfoNCE, or H-InfoNCE) by changing the similarity function.

**Unimodal $p(\mathbf{c}^+ \mid \mathbf{c})$.** We first construct a unimodal conditional, where we expect H-InfoNCE to suffice. We sample $\mathbf{c}^+$ following $c_i^+ \mid \mathbf{c} \sim \mathcal{N}(c_i^+; c_i, \sigma(\mathbf{c})_i^2)$, with $\sigma(\mathbf{c})$ either $0$, isotropic, anisotropic, or heteroscedastic, where $\sigma(\cdot)$ is an affine function followed by `softplus` activation.

Table 1 leads to two main observations. First, models achieve high performance when both their embedding space and model flexibility match the true conditional $p(\mathbf{c}^+ \mid \mathbf{c})$; otherwise we see a decrease in performance, which corroborates the findings of Zimmermann et al. (2021). Notably, we see a clear performance drop with InfoNCE and AnInfoNCE with normalized embedding space. H-InfoNCE improves the performance by a large margin; we explain this with Proposition F.1 and show that heteroscedascity is almost unavoidable. Second, while latent correlations help all models perform well on in-distribution data $p(\mathbf{z})$, only flexible models generalize OOD. Under heteroscedastic noise, the encoders learned with InfoNCE and AnInfoNCE fall short, even trailing the identity function. Interestingly, when $\mathrm{Var}(\mathbf{c}^+ \mid \mathbf{c}) = 0$, generalization performance of InfoNCE is

Table 3: Linear $F_1$ scores on representations (encoder output) and embeddings (projector output) trained on CelebA, under weak or strong augmentations. AdaSSL+GT, a soft performance upper bound, uses the ground-truth attribute difference as $\mathbf{r}$. "——ǁ——" denotes "same as above".

| Model | Pairing | WEAK AUGMENTATION | | STRONG AUGMENTATION | |
|---|---|---|---|---|---|
| | | Repr. | Emb. | Repr. | Emb. |
| InfoNCE | Standard | $0.2698_{\pm 0.0030}$ | $0.1295_{\pm 0.0051}$ | $\underline{0.5965}_{\pm 0.0004}$ | $\underline{0.5694}_{\pm 0.0011}$ |
| InfoNCE | Natural | $0.5473_{\pm 0.0027}$ | $0.3747_{\pm 0.0051}$ | $0.5784_{\pm 0.0008}$ | $0.4941_{\pm 0.0035}$ |
| AnInfoNCE | ——ǁ—— | $0.5413_{\pm 0.0010}$ | $0.4249_{\pm 0.0032}$ | $0.5789_{\pm 0.0008}$ | $0.4987_{\pm 0.0033}$ |
| AdaSSL-V | ——ǁ—— | $\mathbf{0.5784}_{\pm 0.0025}$ | $\mathbf{0.4794}_{\pm 0.0015}$ | $\mathbf{0.6014}_{\pm 0.0008}$ | $\mathbf{0.5706}_{\pm 0.0034}$ |
| AdaSSL-S | ——ǁ—— | $\underline{0.5676}_{\pm 0.0049}$ | $\underline{0.4581}_{\pm 0.0016}$ | $0.5911_{\pm 0.0014}$ | $0.5654_{\pm 0.0007}$ |
| AdaSSL+GT | ——ǁ—— | $0.6818_{\pm 0.0011}$ | $0.6840_{\pm 0.0019}$ | $0.6779_{\pm 0.0003}$ | $0.6832_{\pm 0.0011}$ |

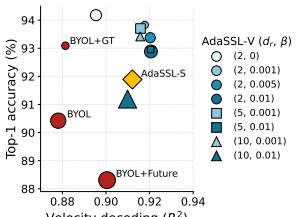

Figure 1: Performance of representations on stochastic Moving-MNIST. Marker size indicates standard deviation.

weaker than the best models in each block, supporting our hypothesis that naturally varying pairs help generalization.

**Complex $p(\mathbf{c}^+ \mid \mathbf{c})$.** In this experiment, we design a DGP where $p(\mathbf{c}^+ \mid \mathbf{c})$ is both multimodal and heteroscedastic. We hypothesize that natural pairs usually differ sparsely in the latent factors, and the differed factors are sometimes conditioned on a latent variable. Therefore, we randomly select some dimensions of $\mathbf{c}^+$ and $\mathbf{c}$ to be shared, while the rest follow Gaussians conditioned on a latent variable $\boldsymbol{\kappa}$, i.e., $c_i, c_i^+ \mid \boldsymbol{\kappa} \sim \mathcal{N}\big(\mu(\boldsymbol{\kappa})_i, \sigma(\boldsymbol{\kappa})_i^2\big)$. See §G.2 for details.

Table 2 shows that InfoNCE and AnInfoNCE are unable to recover latent factors on OOD data. H-InfoNCE improves performance, and AdaSSL variants improve further. We visualize the learned conditionals in §H.1, where AdaSSL best fits the ground truth, suggesting that its improvement comes from more accurate conditional modeling.

### 3.3 Natural images and videos

**Natural images.** Although we do not have access to the ground-truth data generating factors of natural images, we perform experiments on the CelebA dataset (Liu et al., 2015) which contains celebrity images with annotated facial attributes. Beside using *standard pairs* that are augmented versions of the same image, we obtain real-world *natural pairs* by matching different photos of the same celebrity, which differ sparsely in their facial attributes (§G.4). We then train models on paired images and evaluate with affine probes on 40 facial attributes of unseen identities, inducing a natural distribution shift. Results in Table 3 show that standard pairing rely on strong augmentations to work well. However, using natural pairs largely reduces the gap, and only AdaSSL-V consistently improves upon the standard pairing baselines. This exposes InfoNCE's weakness to complex conditionals from natural pairs. We still observe a gap between AdaSSL and AdaSSL+GT, indicating room for improvement in future work.

**World modeling on videos.** In sections above, we have shown AdaSSL models $p(\mathbf{z}^+ \mid \mathbf{z})$ well. Since modeling this transition distribution is central to world modeling on videos, we test AdaSSL on it. We hypothesize that inability to model uncertainty drives the model to discard variant factors. We introduce uncertainty by injecting random changes in velocity between two segments of Moving-MNIST (Srivastava et al., 2015; Drozdov et al., 2024), which are then used as positive pairs. We use BYOL (Grill et al., 2020) as the SSL method for this experiment, whose predictions can condition on a future segment (BYOL+Future) similar to Liu et al. (2025) or the ground-truth change in velocity (BYOL+GT). Fig. 1 shows that AdaSSL captures both the invariant factor, digit, and the variation factor, velocity, better than baselines. Ablation on AdaSSL-V shows its robustness to the dimensionality of $\mathbf{r}$ under proper regularization. We include full results in Table 6 and details in §G.

## 4 Conclusion

In this work, we reveal the limitation of SSL methods when trained on naturally paired data and introduce AdaSSL, which learns a latent variable that captures the uncertainty between pairs. Our approach consistently outperforms existing methods across all benchmarks. We believe this is a promising step in expanding the capability of SSL methods, leading to potentially fruitful advancements in learning generalizable representations, identifiability of high-dimensional images, and world modeling with uncertainty.

**Acknowledgments**

We appreciate the constructive feedback from the anonymous reviewers. We also thank Siddarth Venkatraman, Michael Chong Wang, Emiliano Penaloza, and Omar Salemohamed for insightful discussions, Anirudh Buvanesh and Lucas Maes for precise pointers to literature, and Mehran Shakerinava for proofreading. Additionally, YZ would like to thank Xiaofeng Zhang and Dhanya Sridhar for helpful feedback during the early development of the idea. LC and YZ acknowledge the generous support of the CIFAR AI Chair program. YZ is also supported by the AI Scholarship from Université de Montréal. HG is supported by the UNIQUE Centre (unique.quebec). This research was enabled in part by compute resources provided by Mila (mila.quebec) and the Digital Research Alliance of Canada (alliancecan.ca).

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

# A  Preliminaries

## A.1  Preliminaries: contrastive SSL

Contrastive SSL methods assumes the content factors $\mathbf{c}$ to be roughly unperturbed under the conditional law $p_{Z^+|Z}$, and use an objective that encourage $f(\mathbf{x})$ and $f(\mathbf{x}^+)$ to be similar. To prevent representation collapse where $f$ becomes a constant function, contrastive objectives use another term to encourage the representations to have high entropy (Chen et al., 2020a; Zbontar et al., 2021; Bardes et al., 2022; Wang & Isola, 2020). In this work, we focus on sample-contrastive methods based on InfoNCE (Oord et al., 2018; Chen et al., 2020a), and observe the duality between dimension- and sample-contrastive methods (Garrido et al., 2023a; Balestriero & LeCun, 2022).

The InfoNCE loss has the form:

$$\mathcal{L}_{\text{InfoNCE}} = \mathbb{E}_{\{(\boldsymbol{x}_i,\boldsymbol{x}_i^+)\}_{i=1}^K \overset{\text{iid}}{\sim} p(\mathbf{x},\mathbf{x}^+)} \left[ \frac{1}{K} \sum_{i=1}^K -\log \frac{e^{s(\boldsymbol{x}_i,\boldsymbol{x}_i^+)/\tau}}{\frac{1}{K}\sum_{j=1}^K e^{s(\boldsymbol{x}_i,\boldsymbol{x}_j^+)/\tau}} \right], \tag{6}$$

where $\tau$ is a temperature parameter and $s(\cdot,\cdot)$ is a similarity function over pairs. Intuitively, InfoNCE encourages the similarity function to assign a high score for positive pairs and a low score for pairs that does not come from the true joint. The similarity function often adopts a simple form on the normalized embeddings, i.e., $s(\boldsymbol{x},\boldsymbol{y}) = \psi(\boldsymbol{x})^\top \psi(\boldsymbol{y})$ where $\psi(\cdot) = \frac{f(\cdot)}{\|f(\cdot)\|_2}$. The simplicity of the similarity function allows features to be easily extracted from the embedding space because they are used to discriminate between data points linearly during training (Tschannen et al., 2020).

It has been shown that when the marginal $p(\mathbf{z}^+)$ is uniform, the similarity function implicitly models the log conditional: $s^\star(\mathbf{x},\mathbf{x}^+) \propto \log p(\mathbf{z}^+ \mid \mathbf{z})$ (Zimmermann et al., 2021). With a dot-product similarity, the hypothesis class of $p$ reduces to von Mises-Fisher (vMF) distributions, where $\tau$ controls the concentration strength. Since vMF distribution does not account for anisotropic noise, Rusak et al. (2025) introduces a diagonal matrix $\boldsymbol{\Lambda}$ that weighs the concentration along each dimension: $s(\boldsymbol{x},\boldsymbol{y}) = -(\psi(\boldsymbol{x})-\psi(\boldsymbol{y}))^\top \boldsymbol{\Lambda}(\psi(\boldsymbol{x})-\psi(\boldsymbol{y}))$. **Nevertheless, it remains unclear how to flexibly model an arbitrary conditional distribution $p(\mathbf{z}^+ \mid \mathbf{z})$ while keeping the similarity function simple enough to allow efficient feature extraction.**

## A.2  Preliminaries: non-contrastive SSL

Non-contrastive (or predictive) SSL methods are appealing because they avoid the explicit regularization to prevent representation collapse. Our work addresses the limitations of the invariance component of the SSL objective, making it applicable to these methods as well. Typically, they use asymmetric encoders: an online branch predicts target representations, with a stop-gradient on the target (Grill et al., 2020; Chen & He, 2021). While empirically effective, the reason these design choices prevent collapse is not fully understood (Tian et al., 2021; Zhang et al., 2022; Zhuo et al., 2023). We illustrate our findings with BYOL (Grill et al., 2020), the backbone of many recent successful predictive methods (Guo et al., 2022; Assran et al., 2025):

$$\mathcal{L}_{\text{BYOL}} = \left\| t(\psi(\mathbf{x})) - \psi_{\text{EMA}}(\mathbf{x}^+) \right\|_2^2, \tag{7}$$

where $\psi_{\text{EMA}}$ is the exponential moving average of $\psi$ and $t(\cdot)$ is an MLP predictor.

Because of the predictor, non-contrastive methods frame the problem as predictive learning more explicitly than contrastive ones. **Intuitively, the predictor accounts for cases where $\mathbb{E}[\mathbf{z}^+ \mid \mathbf{z}] \neq \mathbf{z}$; but it remains unclear how it can capture complex conditionals $p(\mathbf{z}^+ \mid \mathbf{z})$—which may be heteroscedastic or even multimodal—without conditioning on additional information.**

# B  Method details

We now present our method, which addresses the aforementioned challenges by modeling uncertainty with a latent variable model. In §2.2, we introduce our overall objective. We then discuss two variants of AdaSSL in §B.2 and §B.3, which optimize this objective in distinct ways.

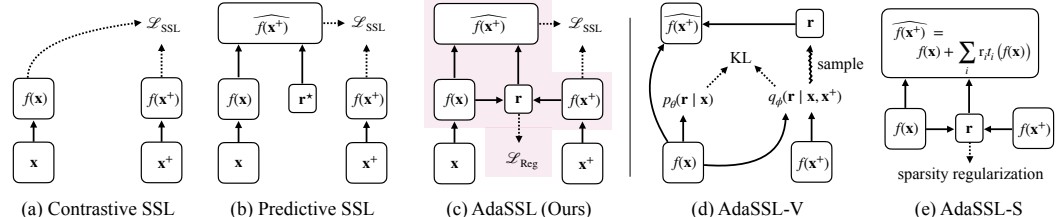

Figure 2: Visual comparison of models. Boxes denote vectors and arrows denote functions. The encoders may not use the same parameters; we use $f$ to denote both for brevity. (a) Contrastive SSL uses a symmetric architecture. (b) Predictive SSL uses a predictor to predict the embeddings of one branch from the other, optionally with the help of some supervision $\mathbf{r}^\star$ related to the difference between the inputs. (c) Our method, AdaSSL, extends predictive SSL by modeling the latent variable $\mathbf{r}$ in the highlighted part. (d) AdaSSL-V learns a variational distribution, $q_\phi(\mathbf{r} \mid \mathbf{x}, \mathbf{x}^+)$, and uses an MLP as predictor. (e) AdaSSl-S regularizes the sparsity of $\mathbf{r}$ and uses a modular predictor.

## B.1  Modeling complex conditionals with a latent variable

In Fig. 2, we visually compare our method to existing approaches. We use a latent variable $\mathbf{r}$ to capture the uncertainty in complex conditional distributions $p(\mathbf{x}^+ \mid \mathbf{x})$, a pushforward of $p(\mathbf{z}^+ \mid \mathbf{z})$ through $g$. The latent variable $\mathbf{r}$ should contain information about $\mathbf{x}^+$ that cannot be solely predicted from $\mathbf{x}$. For example, if $\mathbf{x}$ shows an object just before it passes behind a wall and $\mathbf{x}^+$ shows it after reappearing, $\mathbf{r}$ may represent its acceleration while occluded.

Learning a representation that maximally preserves the mutual information (MI) between paired embeddings is useful for representation learning (Linsker, 1988; Tschannen et al., 2020; Oord et al., 2018). It also provides a way to interpret the desirable properties of $\mathbf{r}$. Specifically, by the chain rule of MI,

$$I(\mathbf{x}; \mathbf{x}^+) = I(\mathbf{x}, \mathbf{r}; \mathbf{x}^+) - I(\mathbf{r}; \mathbf{x}^+ \mid \mathbf{x}). \tag{8}$$

Intuitively, $\mathbf{r}$ should help $\mathbf{x}$ predict $\mathbf{x}^+$ without simply copying $\mathbf{x}^+$. This motivates the general form of our objective:

$$\mathcal{L}_{\text{AdaSSL}} = \mathcal{L}_{\text{SSL}}((\mathbf{x}, \mathbf{r}), \mathbf{x}^+) + \beta \mathcal{L}_{\text{Reg}}(\mathbf{r}), \tag{9}$$

where the SSL term is any standard SSL loss (e.g., $\mathcal{L}_{\text{InfoNCE}}$) that encourages $\mathbf{r}$ to aid prediction of $\mathbf{x}^+$ while the regularizer penalizes $\mathbf{r}$ from becoming an unrestricted shortcut. The hyperparameter $\beta$ controls the strength of regularization per standard practice (Higgins et al., 2017; Locatello et al., 2020). This objective matches the conceptual framework depicted in Fig. 13 of LeCun (2022).

## B.2  AdaSSL-V and a lower bound on $I(\mathbf{x}, \mathbf{x}^+)$

We first learn the posterior $p(\mathbf{r} \mid \mathbf{x}, \mathbf{x}^+)$ with a variational distribution $q_\phi(\mathbf{r} \mid \mathbf{x}, \mathbf{x}^+)$ (Kingma & Welling, 2014; Sohn et al., 2015). The joint then becomes $\tilde{p}(\mathbf{x}, \mathbf{x}^+, \mathbf{r}) \coloneqq p(\mathbf{x}, \mathbf{x}^+)q(\mathbf{r} \mid \mathbf{x}, \mathbf{x}^+)$. The informational-theoretical properties of contrastive learning allow us to optimize a lower bound on $I(\mathbf{x}, \mathbf{x}^+)$[5]. Specifically, the first term in Eq. 8 is bounded by InfoNCE (Oord et al., 2018) by treating $(\mathbf{x}, \mathbf{r})$ as a single variable:

$$I_{\tilde{p}}(\mathbf{x}, \mathbf{r}; \mathbf{x}^+) \geq -\mathcal{L}_{\text{InfoNCE}} = \mathbb{E}_{\{(\boldsymbol{x}_i, \boldsymbol{x}_i^+, \boldsymbol{r}_i)\}_{i=1}^K \overset{\text{iid}}{\sim} \tilde{p}} \left[ \frac{1}{K} \sum_{i=1}^K \log \frac{e^{s(\boldsymbol{x}_i, \boldsymbol{x}_i^+, \boldsymbol{r}_i)/\tau}}{\frac{1}{K} \sum_{j=1}^K e^{s(\boldsymbol{x}_i, \boldsymbol{x}_j^+, \boldsymbol{r}_i)/\tau}} \right]. \tag{10}$$

We derive a bound for the second term in §E:

$$-I_{\tilde{p}}(\mathbf{r}; \mathbf{x}^+ \mid \mathbf{x}) \geq -\mathcal{L}_{\text{Reg}} = -\mathbb{E}_{p(\mathbf{x}, \mathbf{x}^+)} \left[ D_{\text{KL}}(q_\phi(\mathbf{r} \mid \mathbf{x}, \mathbf{x}^+) \| p_\theta(\mathbf{r} \mid \mathbf{x})) \right]. \tag{11}$$

Thus, by introducing a latent posterior, we obtain a tractable lower bound on $I(\mathbf{x}; \mathbf{x}^+)$. In practice, we parameterize $q_\phi$ and $p_\theta$ using lightweight MLPs on top of the embeddings $f(\mathbf{x})$ and $f(\mathbf{x}^+)$, modeling both as factorized Gaussians. Plugging the terms into Eq. 9, we get

$$\mathcal{L}_{\text{AdaSSL-V}} = \mathcal{L}_{\text{SSL}}\big(\mathbb{E}_{q_\phi}\psi_1(\mathbf{x}, \mathbf{r}), \psi_2(\mathbf{x}^+)\big) + \beta D_{\text{KL}}(q_\phi(\mathbf{r} \mid \mathbf{x}, \mathbf{x}^+) \| p_\theta(\mathbf{r} \mid \mathbf{x})). \tag{12}$$

---

[5]One can equivalently replace $I(\mathbf{x}; \mathbf{x}^+)$ with $I(f(\mathbf{x}); f(\mathbf{x}^+))$, since our method operates on paired embeddings. For simplicity, we use the notation $I(\mathbf{x}; \mathbf{x}^+)$ throughout, but in practice our method aims to maximize $I(f(\mathbf{x}); f(\mathbf{x}^+)) \leq I(\mathbf{x}; \mathbf{x}^+)$.

Table 4: Identifiability results on 3DIdent. AdaSSL achieves the best disentanglement and $R^2$ scores. "—ꞮꞮ—" denotes "same as above".

| Model | Pairing | DCI disent. ($\uparrow$) | $R^2$ ($\uparrow$) |
|---|---|---|---|
| $\beta$-VAE$_{\beta=1}$ | - | $0.2076 \pm 0.0243$ | $0.6649 \pm 0.0307$ |
| $\beta$-VAE$_{\beta=16}$ | - | $0.1883 \pm 0.0191$ | $0.6672 \pm 0.0216$ |
| $\beta$-VAE$_{\beta=100}$ | - | $0.3352 \pm 0.0468$ | $0.6691 \pm 0.0342$ |
| AdaGVAE$_{\beta=1}$ | Natural | $0.4098 \pm 0.0413$ | $0.6436 \pm 0.0343$ |
| AdaGVAE$_{\beta=16}$ | —ꞮꞮ— | $0.3800 \pm 0.0131$ | $0.6511 \pm 0.0141$ |
| AdaGVAE$_{\beta=100}$ | —ꞮꞮ— | $\underline{0.4582} \pm 0.0154$ | $0.6213 \pm 0.0143$ |
| InfoNCE | Standard | $0.1447 \pm 0.0032$ | $0.3382 \pm 0.0074$ |
| AnInfoNCE | —ꞮꞮ— | $0.1349 \pm 0.0007$ | $0.3704 \pm 0.0113$ |
| InfoNCE | Natural | $0.1178 \pm 0.0073$ | $0.8184 \pm 0.0047$ |
| AnInfoNCE | —ꞮꞮ— | $0.2772 \pm 0.0184$ | $0.8243 \pm 0.0002$ |
| AdaSSL-V$_{\text{Additive}}$ | —ꞮꞮ— | $\mathbf{0.4661} \pm 0.0467$ | $0.8857 \pm 0.0012$ |
| AdaSSL-V$_{\text{Linear}}$ | —ꞮꞮ— | $0.2756 \pm 0.0266$ | $\mathbf{0.9331} \pm 0.0077$ |
| AdaSSL-V$_{\text{MLP}}$ | —ꞮꞮ— | $0.1027 \pm 0.0048$ | $0.8948 \pm 0.0017$ |
| AdaSSL-S | —ꞮꞮ— | $0.1777 \pm 0.1009$ | $\underline{0.9309} \pm 0.0096$ |

Figure 3: AdaSSL-V performs controllable retrieval. From the query image $\mathbf{x}$, we sample from different dimensions of the learned prior $p_\theta(\tilde{\mathbf{r}} \mid \mathbf{x})$ which correspond to interpretable changes in the edited image $t(f(\mathbf{x}), \tilde{\mathbf{r}})$.

We call this variant of our method **AdaSSL-V**(variational).

*Remark.* Although AdaSSL-V is only theoretically justified for contrastive SSL, one can use a non-constrastive SSL loss as well because they still encourage $\mathbf{r}$ to aid prediction of $\mathbf{x}^+$.

**Similarity function.** As discussed in §A.1, our goal is to have a similarity function that is flexible yet simple. With $\mathbf{r}$ as a latent variable, we still use the dot-product similarity on embeddings:

$$s(\mathbf{x}, \mathbf{x}^+, \mathbf{r}) = \psi_1(\mathbf{x}, \mathbf{r})^\top \psi_2(\mathbf{x}^+), \text{ where } \psi_1(\mathbf{x}, \mathbf{r}) = \frac{t(f(\mathbf{x}), \mathbf{r})}{\|t(f(\mathbf{x}), \mathbf{r})\|_2}, \psi_2(\mathbf{x}^+) = \frac{f(\mathbf{x}^+)}{\|f(\mathbf{x}^+)\|_2}. \tag{13}$$

Specifically, we *edit* $f(\mathbf{x})$ with the help of $\mathbf{r}$ and an editing function $t(\cdot, \cdot)$ such that it lies in the vicinity of $f(\mathbf{x}^+)$. For InfoNCE, we parameterize $t$ with a linear projection or two-layer MLPs. For BYOL, we directly use $\mathbf{r}$ as an additional input to its predictor.

### B.3 AdaSSL-S and sparse modular edits

Natural transitions usually correspond to sparse changes in the latent factors, an inductive bias widely adopted in the identifiability literature (Ahuja et al., 2022; Klindt et al., 2021; Lippe et al., 2023). Therefore, we hypothesize that we can implement Eq. 9 by predicting $\mathbf{r}$ and regularizing its sparsity. **AdaSSL-S**(parse) realizes this idea. Instead of learning a variational posterior, we predict $\mathbf{r}$ deterministically from $f(\mathbf{x})$ and $f(\mathbf{x}^+)$: $\mathbf{r} = m(f(\mathbf{x}), f(\mathbf{x}^+))$, where $m$ is an MLP followed by `tanh` activation. We then regularize the sparsity of $\mathbf{r}$:

$$\mathcal{L}_{\text{AdaSSL-S}} = \mathcal{L}_{\text{SSL}}\big(\psi_1(\mathbf{x}, \mathbf{r}), \psi_2(\mathbf{x}^+)\big) + \beta\|\mathbf{r}\|_0, \tag{14}$$

where the $L_0$ penalty is made differentiable through the Gumbel-Sigmoid estimator similar to the one used by Lachapelle et al. (2022); Brouillard et al. (2020).

Inspired by Ibrahim et al. (2022); Hu et al. (2022), we use a modular editing function $t$:

$$t(f(\mathbf{x}), \mathbf{r}) = f(\mathbf{x}) + \sum_{i=1}^{d_r} \mathbf{r}_i t_i(f(\mathbf{x})) = f(\mathbf{x}) + \sum_{i=1}^{d_r} \mathbf{r}_i (\mathbf{B}_i \mathbf{A}_i f(\mathbf{x}) + b_i), \tag{15}$$

where $d_r$ is the dimensionality of $\mathbf{r}$. Each editing function $t_i(\cdot)$ is an affine transformation parameterized by a rank-1 matrix $\mathbf{B}_i \mathbf{A}_i$ and a scalar offset $b_i$. This design is motivated by the assumption that differences between the paired embeddings lie in a low-dimensional latent subspace, where edits are applied.

Similarly to AdaSSL-V, AdaSSL-S is applicable to both contrastive and non-contrastive SSL.

## C Causal representation learning

In this section, we show AdaSSL can be used to recover *all* data generating factors from natural pairs on 3DIdent (Zimmermann et al., 2021), a dataset of realistically rendered images of a teapot

varying in ten data generating factors such as position, spotlight, and hue. Following Locatello et al. (2020), we generate natural pairs by first drawing two samples from the marginal latent distribution. Then, each latent coordinate is replaced with some probability by the corresponding coordinate from the other sample. We evaluate (a) disentanglement in the learned embeddings with the DCI disentanglement score (Eastwood & Williams, 2018), and (b) identifiability up to affine transformations with $R^2$.

In this experiment, we focus on contrastive SSL, but also compare with classic disentanglement methods, including $\beta$-VAE (Higgins et al., 2017) and AdaGVAE (Locatello et al., 2020). Table 4 shows that $\beta$-VAE and AdaGVAE fail to identify the latent factors, though AdaGVAE achieves decent disentanglement. InfoNCE with augmentations performs worse, likely because augmentation invariance conflicts with identifiability. SSL baselines using natural pairs achieve good identifiability but yield more entangled latent factors. We hypothesize that AdaSSL's regularization encourages efficient encodings of $\mathbf{r}$, akin to $\beta$-VAE (Higgins et al., 2017; Burgess et al., 2018). Since $\mathbf{r}$ is modeled as factorized Gaussians, some disentanglement in $\mathbf{r}$ is expected. To verify this, we vary the complexity of the editing function $t$ (additive, linear, nonlinear), as shown in the subscripts in Table 4. Indeed, simpler $t$ leads to more disentagled embeddings while consistently outperforming baselines on regression performance. In particular, AdaSSL-V$_{\text{Linear}}$ and AdaSSL-S, which both use linear editing functions, achieve the highest identifiability.

To better understand the learned $\mathbf{r}$, we visualize its effect by retrieving nearest neighbor of a query image $\mathbf{x}$ after editing it with samples $\tilde{\mathbf{r}}$ (Fig. 3). Given evidence of disentanglement, we expect sampling specific latent dimensions to induce meaningful changes in the edited embeddings $t(f(\mathbf{x}), \tilde{\mathbf{r}})$. Concretely, we sample $\tilde{r}_i \sim p_\theta(r_i \mid \mathbf{x})$ for $i \in \mathbb{L} \subseteq [d_r]$ for some set of latent indices $\mathbb{L}$, and fix all others to their expectations. Fig. 3 shows results for three different $\mathbb{L}$'s. We find that we can retrieve objects that differ in position, spotlight, and color, while leaving most other factors unchanged, though orientation remains entangled with other factors. Finally, when sampling from the full prior, we retrieve images that differ sparsely in latent factors, consistent with the training DGP.

Together, these results highlight SSL as a promising path for CRL for its efficiency (no reconstruction) and demonstrated scalability to high-dimensional images.

# D    Related work

**Self-supervised learning.**    SSL in the latent space has evolved from solving hand-crafted *pretext tasks* (Noroozi & Favaro, 2016; Doersch et al., 2015; Dosovitskiy et al., 2014; Gidaris et al., 2018) to learning semantic-preserving representations from invariance to augmentations (Oord et al., 2018; Wu et al., 2018; Gutmann & Hyvärinen, 2010; Chen et al., 2020b; Caron et al., 2020; Wu et al., 2018; He et al., 2020; Radford et al., 2021; Caron et al., 2021; Zbontar et al., 2021; Bardes et al., 2022; Ermolov et al., 2021; Chen & He, 2021; Grill et al., 2020; Assran et al., 2023; Baevski et al., 2022; Caron et al., 2020; He et al., 2016). Studies have also explored the relationship between invariant representations and variational inference (Bizeul et al., 2024; Sinha & Dieng, 2021). Beyond invariance, equivariant representations preserve transformation information (Hinton et al., 2011). In SSL, this is achieved by providing augmentation parameters to the predictor (Garrido et al., 2023b; Ghaemi et al., 2024; Devillers & Lefort, 2023; Garrido et al., 2024; Park et al., 2022), or using subspaces for different invariances (Xiao et al., 2021; Eastwood et al., 2023). However, these approaches are tied to chosen augmentations and break down when the sources of uncertainty are unknown. Alternatively, one can exploit the invariance between observation pairs that are transformed similarly (Shakerinava et al., 2022), or model transformation with Lie groups (Ibrahim et al., 2022); the latter requires jointly optimizing the vanilla SSL loss and only learns a single factor of variation. Lastly, Lavoie et al. (2024) reduce prediction uncertainty between image–caption pairs by conditioning visual representations on textual ones through a cross-attention mechanism, thereby improving the feature diversity of contrastive vision–language models. Unlike prior work, our method does not require transformation labels, handles multiple varying factors, and provides a simple, theoretically justified objective that is compatible with standard SSL methods across diverse settings.

**Causal representation learning.**    Much research examines recovering data-generating factors and their causal relations (Hyvarinen & Morioka, 2016; Schölkopf et al., 2021; von Kügelgen et al., 2023; Ahuja et al., 2023; Brehmer et al., 2022; Locatello et al., 2020; Lachapelle et al., 2022; Lippe et al., 2023; Klindt et al., 2021; Ahuja et al., 2022; Lippe et al., 2022; Yao et al., 2025). While

offering theoretical guarantees, these methods often rely on strong assumptions or probabilistic generative models, limiting scalability. SSL has been connected to CRL (Zimmermann et al., 2021; Kügelgen et al., 2021; Rusak et al., 2025; Yao et al., 2024), where studies focus on identifying the content factors that follow simple conditionals (§A.1). This work relaxes these assumptions by allowing structured variation between paired latents and demonstrates strong performance on weakly-supervised CRL, a step towards understanding and advancing SSL (Reizinger et al., 2025).

**World modeling with SSL.** Unlike image-based SSL that rely on augmentations, video world models with SSL learn the transition dynamics of videos, often by predicting target frames given some context (Sermanet et al., 2018; Feichtenhofer et al., 2021; Bardes et al., 2024; Assran et al., 2025; Schwarzer et al., 2021; Guo et al., 2022). Through the process, the model learns useful representations for downstream tasks such as video understanding. A key challenge is that uncertainty grows with the temporal gap between positive pairs, forcing models to fix temporal resolution (Feichtenhofer et al., 2021; Bardes et al., 2024), which may limit their ability to learn features at different levels of abstractions (Zacks & Tversky, 2001) because the model can discard variant factors. Introducing a latent variable $\mathbf{r}$, as we do, can reduce the uncertainty and learn more diverse features (§3.3). Finally, although we focus on improving SSL that does not require reconstruction, we note there are successful approaches that predict in the observation space (Schmidt & Jiang, 2024; Tong et al., 2022; Feichtenhofer et al., 2022; Jang et al., 2024; Bruce et al., 2024; Yang et al., 2024).

# E Derivation of Eq. 11

$$- I_{\tilde{p}}(\mathbf{r}; \mathbf{x}^+ \mid \mathbf{x})$$

$$= -\mathbb{E}_{\tilde{p}}\left[\log \frac{q(\mathbf{r} \mid \mathbf{x}, \mathbf{x}^+)}{\tilde{p}(\mathbf{r} \mid \mathbf{x})}\right]$$

$$= -\mathbb{E}_{\tilde{p}}\left[\log \frac{q(\mathbf{r} \mid \mathbf{x}, \mathbf{x}^+)}{\tilde{p}(\mathbf{r} \mid \mathbf{x})} + \log p(\mathbf{r} \mid \mathbf{x}) - \log p(\mathbf{r} \mid \mathbf{x})\right]$$

$$= -\mathbb{E}_{\tilde{p}}\left[\log \frac{q(\mathbf{r} \mid \mathbf{x}, \mathbf{x}^+)}{p(\mathbf{r} \mid \mathbf{x})} + \log \frac{p(\mathbf{r} \mid \mathbf{x})}{\tilde{p}(\mathbf{r} \mid \mathbf{x})}\right]$$

$$= -\mathbb{E}_{p(\mathbf{x},\mathbf{x}^+)}\left[D_{\mathrm{KL}}(q(\mathbf{r} \mid \mathbf{x}, \mathbf{x}^+)\|p(\mathbf{r} \mid \mathbf{x}))\right] + \mathbb{E}_{\tilde{p}}\left[\log \frac{\tilde{p}(\mathbf{r} \mid \mathbf{x})}{p(\mathbf{r} \mid \mathbf{x})}\right]$$

$$= -\mathbb{E}_{p(\mathbf{x},\mathbf{x}^+)}\left[D_{\mathrm{KL}}(q(\mathbf{r} \mid \mathbf{x}, \mathbf{x}^+)\|p(\mathbf{r} \mid \mathbf{x}))\right] + \int\int\int p(\mathbf{x})p(\mathbf{x}^+ \mid \mathbf{x})q(\mathbf{r} \mid \mathbf{x}, \mathbf{x}^+) \log \frac{\tilde{p}(\mathbf{r} \mid \mathbf{x})}{p(\mathbf{r} \mid \mathbf{x})} d\mathbf{r}d\mathbf{x}^+d\mathbf{x}$$

$$= -\mathbb{E}_{p(\mathbf{x},\mathbf{x}^+)}\left[D_{\mathrm{KL}}(q(\mathbf{r} \mid \mathbf{x}, \mathbf{x}^+)\|p(\mathbf{r} \mid \mathbf{x}))\right] + \int\int p(\mathbf{x})\left(\int p(\mathbf{x}^+ \mid \mathbf{x})q(\mathbf{r} \mid \mathbf{x}, \mathbf{x}^+)d\mathbf{x}^+\right) \log \frac{\tilde{p}(\mathbf{r} \mid \mathbf{x})}{p(\mathbf{r} \mid \mathbf{x})} d\mathbf{r}d\mathbf{x}$$

$$= -\mathbb{E}_{p(\mathbf{x},\mathbf{x}^+)}\left[D_{\mathrm{KL}}(q(\mathbf{r} \mid \mathbf{x}, \mathbf{x}^+)\|p(\mathbf{r} \mid \mathbf{x}))\right] + \int\int p(\mathbf{x})\tilde{p}(\mathbf{r} \mid \mathbf{x}) \log \frac{\tilde{p}(\mathbf{r} \mid \mathbf{x})}{p(\mathbf{r} \mid \mathbf{x})} d\mathbf{r}d\mathbf{x}$$

$$= -\mathbb{E}_{p(\mathbf{x},\mathbf{x}^+)}\left[D_{\mathrm{KL}}(q(\mathbf{r} \mid \mathbf{x}, \mathbf{x}^+)\|p(\mathbf{r} \mid \mathbf{x}))\right] + \mathbb{E}_{p(\mathbf{x})\tilde{p}(\mathbf{r}|\mathbf{x})}\left[\log \frac{\tilde{p}(\mathbf{r} \mid \mathbf{x})}{p(\mathbf{r} \mid \mathbf{x})}\right]$$

$$= -\mathbb{E}_{p(\mathbf{x},\mathbf{x}^+)}\left[D_{\mathrm{KL}}(q(\mathbf{r} \mid \mathbf{x}, \mathbf{x}^+)\|p(\mathbf{r} \mid \mathbf{x}))\right] + \mathbb{E}_{p(\mathbf{x})}[D_{\mathrm{KL}}(\tilde{p}(\mathbf{r} \mid \mathbf{x})\|p(\mathbf{r} \mid \mathbf{x}))]$$

$$\geq -\mathbb{E}_{p(\mathbf{x},\mathbf{x}^+)}\left[D_{\mathrm{KL}}(q(\mathbf{r} \mid \mathbf{x}, \mathbf{x}^+)\|p(\mathbf{r} \mid \mathbf{x}))\right].$$

# F Theory

**Lemma F.1.** *Let $A \in \mathbb{R}^{m \times n}$ and let $\Sigma \in \mathbb{R}^{n \times n}$ be symmetric positive definite. Then*

$$\mathrm{range}(A\Sigma A^\top) = \mathrm{range}(A).$$

*Proof.* For any $x \in \mathbb{R}^m$, we have

$$x^\top(A\Sigma A^\top)x = (A^\top x)^\top \Sigma(A^\top x).$$

Since $\Sigma$ is symmetric positive definite, the right-hand side is zero if and only if $A^\top x = 0$. Thus,

$$\ker(A\Sigma A^\top) = \ker(A^\top).$$

Taking orthogonal complements yields

$$\mathrm{range}(A\Sigma A^\top) = \mathrm{range}(A).$$

$\square$

*Remark.* This is a standard linear algebra fact; we include it here for completeness.

**Proposition F.1.** *Let $\mathbb{S}^k \subset \mathbb{R}^{k+1}$ denote the $k$-dimensional unit sphere. Let $g : \mathbb{R}^d \to \mathbb{R}^{d'}$ be $C^1$ diffeomorphic to its image, and let $f : \mathbb{R}^{d'} \to \mathbb{S}^k$ be $C^1$ almost everywhere. Define $h := f \circ g : \mathbb{R}^d \to \mathbb{S}^k$. Assume further that the random vectors $z, z^+ \in \mathbb{R}^d$ are sampled as*

$$z \sim p_Z, \qquad z^+ = z + \varepsilon, \quad \varepsilon \sim p_\varepsilon,$$

*where $p_Z$ is not a point mass and $\varepsilon$ is independent of $z$, $\mathbb{E}[\varepsilon] = 0$, and $\mathrm{Cov}(\varepsilon) \succ 0$.*

*Suppose that for $p_Z$-almost every $z$ we have $h(z) \in \mathbb{S}^k$ and $\mathrm{rank}\, Dh(z) = d$. Write $H = h(z)$ and $H^+ = h(z^+)$. Then the conditional law*

$$p_{H^+|H}(h(z^+) \mid h(z)),$$

*is necessarily heteroscedastic: its conditional variance depends on $h(z)$ for $p_Z$-almost every $z$.*

Proposition F.1 shows that heteroscedasticity between paired embeddings emerges from the geometric mismatch between the embedding space and the ground-truth latent space, regardless of the encoding function or embedding dimensionality. Here, we explicitly show the case of projecting from unbounded latent space $\mathbb{R}^{d_z}$ to normalized embedding space $\mathbb{S}^{d_f}$ and discuss the reverse scenario in Proposition F.2. Consequently, common similarity functions such as the dot product fail to capture this conditional variance, since they aggregate the variability uniformly across all embedding directions and data pairs. We show this empirically in §3.2.

*Proof.* Fix $z$ where $h$ is $C^1$ and $\mathrm{rank}\, Dh(z) = d$. For $\sigma > 0$ small, define $z^+ = z + \sigma\varepsilon$ with $\varepsilon \sim p_\varepsilon$. A first-order Taylor expansion and the delta method give

$$h(z + \sigma\varepsilon) = h(z) + Dh(z)\,\sigma\varepsilon + o(\sigma),$$

which implies

$$\mathrm{Cov}[h(z + \sigma\varepsilon) \mid z] = \sigma^2 Dh(z)\,\Sigma\,Dh(z)^\top + o(\sigma^2).$$

If the conditional covariance were homoscedastic at leading order, there exists a fixed positive semidefinite matrix $C$ such that

$$Dh(z)\,\Sigma\,Dh(z)^\top \equiv C \quad \text{for } p_Z\text{-almost every } z.$$

Let $W := \mathrm{range}(C)$. By Lemma F.1 and $\Sigma \succ 0$ we have

$$\mathrm{range}\big(Dh(z)\big) = \mathrm{range}\big(Dh(z)\Sigma Dh(z)^\top\big) = \mathrm{range}(C) = W,$$

so $\mathrm{range}(Dh(z)) \equiv W$ is the same $d$-dimensional subspace for $p_Z$-almost every $z$. Because $h(z) \in \mathbb{S}^k$ we have $\|h(z)\|^2 \equiv 1$, so differentiating yields

$$h(z)^\top Dh(z) = 0,$$

i.e. $\mathrm{range}(Dh(z)) \subset h(z)^\perp$. Since $\mathrm{range}(Dh(z)) = W$ for almost every $z$, we obtain $W \subset h(z)^\perp$ almost everywhere, hence $h(z) \in W^\perp$ for almost every $z$.

Pick any nonzero $w \in W$. Then $w^\top h(z) = 0$ for almost every $z$, and differentiating gives $w^\top Dh(z) = 0$ for almost every $z$, i.e. $w \perp \mathrm{range}(Dh(z)) = W$. Thus $W \subset W^\perp$, which forces $W = \{0\}$. This contradicts $\mathrm{rank}\, Dh(z) = d > 0$. Therefore the hypothesis that $Dh(z)\Sigma Dh(z)^\top$ is constant in $z$ is false, so the leading-order conditional covariance must depend on $z$ for $p_Z$-almost every $z$. $\square$

*Remark.* The above argument establishes heteroscedasticity at *leading order* in the noise scale $\sigma$, which rigorously shows that the conditional covariance depends on $z$ for sufficiently small $\sigma$. For larger $\sigma$, higher-order terms in the Taylor expansion of $h$ become significant and the exact conditional covariance may be more complicated; nevertheless, the local Jacobian $Dh(z)$ still transforms the noise differently at different points, so the conditional variance remains intuitively location-dependent, even if no simple closed-form expression exists.

**Proposition F.2** (Tangent-space variant of Proposition F.1)**.** *Let $\mathbb{S}^k \subset \mathbb{R}^{k+1}$ denote the k-dimensional unit sphere, and $U \subset \mathbb{S}^k$ an open set. Let $g : \mathbb{S}^k \to \mathbb{S}^{k'}$ be $C^1$ diffeomorphic to its image, and let $f : \mathbb{S}^{k'} \to \mathbb{R}^d$ be $C^1$ almost everywhere. Define $h := f \circ g : U \to \mathbb{R}^d$. We assume that $h$ is nondegenerate, i.e., $h(U)$ is not contained in any proper affine subspace of its intrinsic dimension. Suppose that for almost every $z \in U$, the derivative $Dh(z) : T_z\mathbb{S}^k \to \mathbb{R}^d$ has full rank, i.e. $\operatorname{rank} Dh(z) = k$. Assume further that the conditional distribution of $z^+ \in \mathbb{S}^k$ given $z$ is locally Gaussian in the tangent space*

$$p(z^+ \mid z) \propto \exp\left( -(z^+ - z)^\top \Lambda (z^+ - z) \right),$$

*with a constant positive definite diagonal matrix $\Lambda$.*

*Define $H = h(z)$ and $H^+ = h(z^+)$. Then for generic nondegenerate $C^1$ maps $h$, the conditional law*

$$p_{H^+|H}(h(z^+) \mid h(z)),$$

*is heteroscedastic for almost every $z \in U$.*

*Proof.* We construct $z^+$ by a small Gaussian step in $\mathbb{R}^{k+1}$ and normalization:

$$z^+ = \frac{z + \varepsilon}{\|z + \varepsilon\|}, \quad \varepsilon \sim \mathcal{N}(0, \Lambda^{-1}).$$

A first-order approximation for small $\varepsilon$ gives

$$z^+ - z = P_z \varepsilon + O(\|\varepsilon\|^2),$$

where $P_z = I - zz^\top$ is the projector to the tangent space, and the pushforward density on the sphere matches

$$p(z^+ \mid z) \propto \exp\left( -(z^+ - z)^\top \Lambda (z^+ - z) \right)$$

up to higher-order terms.

Fix $z \in U$ where $h$ is $C^1$ and $\operatorname{rank} Dh(z)$ has full rank. A Euclidean Taylor expansion gives

$$h(z^+) = h(z) + Dh(z)(z^+ - z) + O(\|z^+ - z\|^2).$$

Substituting $z^+ - z \approx P_z \varepsilon$

$$h(z^+) = h(z) + Dh(z)P_z \varepsilon + R(z),$$

where $R(z)$ collects higher-order terms, and the leading-order conditional covariance is

$$\operatorname{Cov}(h(z^+) \mid z) = Dh(z)\Sigma_z^{\text{tan}} Dh(z)^\top + R(z), \quad \Sigma_z^{\text{tan}} = P_z \Lambda^{-1} P_z,$$

with $R(z)$ continuous and symmetric.

Suppose that $\operatorname{Cov}(h(z^+) \mid z)$ were constant across $z \in U$. With $\Sigma_z^{\text{tan}} \succ 0$, the range of the leading term $\operatorname{range}(Dh(z)\Sigma_z^{\text{tan}} Dh(z)^\top) = \operatorname{range}(Dh(z))$ would have to be the same subspace $W \subset \mathbb{R}^d$ for almost every $z \in U$.

For any differentiable curve $z(t) \subset U$ through points where $Dh(z(t))$ has full rank, we can write

$$\frac{d}{dt}h(z(t)) = Dh(z(t))\dot{z}(t) \in W$$

Integrating along all such curves in $U$ gives

$$h(U) \subset h(z_0) + W,$$

for some base point $z_0$. This would imply that the image $h(U)$ is contained in a fixed affine subspace $W \subset R^d$, contradicting the nondegeneracy assumption on $h$. Therefore, a constant pushforward covriance can only occur in the trivial case of no noise ($\Sigma_z^{\text{tan}} = 0$, or $\Lambda^{-1} = 0$) or in a highly specific algebraic cancellation between $Dh(z)$ and $\Sigma_z^{\text{tan}}$. For generic nondegenerate $C^1$ maps $h$ and almost every $z \in U$, the conditional covariance is therefore heteroscedastic. $\square$

*Remark.* This is analogous to Proposition F.1, but with domain and codomain swapped; the argument relies on the Jacobian of the map and the local Gaussian structure in the tangent space.

**Proposition F.3** (Extension of Proposition F.1). *Let $g : \mathbb{R}^d \to \mathbb{R}^{d'}$ be a $C^2$ with a local diffeomorphism and $f : \mathbb{R}^{d'} \to \mathcal{M}$ be $C^2$ almost everywhere. Define $h := f \circ g : \mathbb{R}^d \to \mathcal{M}$ where $\mathcal{M}$ is a Riemannian manifold with strictly positive sectional curvature on a nonempty open set. Assume further that the random vectors $z, z^+ \in \mathbb{R}^d$ are sampled as*

$$z \sim p_Z, \qquad z^+ = z + \varepsilon, \quad \varepsilon \sim p_\varepsilon,$$

*where $p_Z$ is not a point mass and $\varepsilon$ is independent of $z$, $\mathbb{E}[\varepsilon] = 0$, and $\mathrm{Cov}(\varepsilon) \succ 0$.*

*Suppose that for $p_Z$-almost every $z$ we have $h(z) \in \mathcal{M}$ and $\mathrm{rank}\, Dh(z) = d$. Write $H = h(z)$ and $H^+ = h(z^+)$. Then the conditional law*

$$p_{H^+|H}(h(z^+) \mid h(z)),$$

*is necessarily heteroscedastic: its conditional variance depends on $h(z)$ for $p_Z$-almost every $z$.*

*Proof.* Following the same reasoning as in Theorem F.1, homoscedasticity at leading order would require a constant positive semidefinite matrix $C$ such that

$$Dh(z)\, \Sigma\, Dh(z)^\top \equiv C \quad \text{for } p_Z\text{-almost every } z.$$

Since $\Sigma \succ 0$, the above condition is equivalent to requiring that

$$\langle u, v \rangle_\Sigma := u^\top \Sigma v = \langle Dh(z)u, Dh(z)v \rangle_{\mathbb{R}^{k+1}} \quad \forall u, v \in \mathbb{R}^d, \text{ for a.e. } z$$

i.e., $h$ is a local Riemannian isometry from the flat space $(\mathbb{R}^d, \langle \cdot, \cdot \rangle_\Sigma)$ to the positively curved manifold $(\mathcal{M}, g_\mathcal{M})$. However, local isometries preserve sectional curvature (Gauss' Theorema Egregium), so no such local isometry from an open subset of $\mathbb{R}^d$ to an open subset of $\mathcal{M}$ exists. Hence, the homoscedasticity condition cannot hold.

Therefore, for all sufficiently small $\sigma > 0$, the conditional covariance

$$\mathrm{Cov}[h(z + \sigma\varepsilon) \mid z] = \sigma^2 Dh(z)\, \Sigma\, Dh(z)^\top + o(\sigma^2)$$

depends on $z$, and the conditional distribution of $h(z^+)$ given $h(z)$ is necessarily heteroscedastic for $p_Z$-almost every $z$. $\qquad \square$

# G Implementation details

## G.1 Leveraging an additional view

For both AdaSSL-V and AdaSSL-S, we expect the model to learn what explains the differences in the paired views in $\mathbf{r}$. However, if our goal is to encode $\mathbf{c}$ and learn a representation invariant to $\mathbf{s}$ (§2.1), we might not want to encode $\mathbf{s}$ and should prioritize learning $\mathbf{c}$. For example, invariance to certain style factors is crucial for generalization (Deng et al., 2022) and preventing shortcut solutions in SSL (Chen et al., 2020a).

One way to ensure $\mathbf{r}$ learns the right directions is to use a surrogate view $\mathbf{x}^{++}$—whose relationship with $\mathbf{x}$ in the underlying content factors $\mathbf{c}$ and $\mathbf{c}^{++}$ mimic that between $\mathbf{x}^+$ and $\mathbf{x}$—to replace $\mathbf{x}^+$. In other words, AdaSSL-V uses $\mathbf{r}$ sampled from $q_\phi(\mathbf{r} \mid f(\mathbf{x}), f(\mathbf{x}^{++}))$ and AdaSSL-S uses $\mathbf{r}$ predicted by $m(f(\mathbf{x}), f(\mathbf{x}^{++}))$. These additional views are usually easy to obtain, e.g., by augmentations. We describe the $\mathbf{x}^{++}$ that we use in each experiment below.

It is crucial to note that our method does not depend on the presence of the additional view. When we want to learn *all* the data generating factors, i.e., when $\mathbf{c} = \mathbf{z}$, we do not use additional views (§C).

## G.2 Numerical experiments in §3.2

In the numerical experiments, most of our setup follows prior work (Kügelgen et al., 2021; Zimmermann et al., 2021). We list the similarity functions used by the models in Table 5.

Table 5: Similarity functions used by different models, where $\psi(\cdot) = \frac{f(\cdot)}{\|f(\cdot)\|_2}$ if the model assumes a normalized latent space, in which case InfoNCE and AdaSSL's similarity functions are equivalent to a dot product; otherwise $\psi(\cdot) = f(\cdot)$. The same applies to $\psi_1$ and $\psi_2$, whose subscripts are used to indicate the asymmetry of H-InfoNCE and AdaSSL. Note that in Table 1, H-InfoNCE has $\psi_1 = \psi_2$ because $\mathbb{E}[\mathbf{c}^+ \mid \mathbf{c}] = \mathbf{c}$.

| Model | $s(\boldsymbol{x}, \boldsymbol{y})$ |
|---|---|
| InfoNCE | $-\lambda(\psi(\boldsymbol{x}) - \psi(\boldsymbol{y}))^\top (\psi(\boldsymbol{x}) - \psi(\boldsymbol{y}))$ |
| AnInfoNCE | $-(\psi(\boldsymbol{x}) - \psi(\boldsymbol{y}))^\top \boldsymbol{\Lambda} (\psi(\boldsymbol{x}) - \psi(\boldsymbol{y}))$ |
| H-InfoNCE | $-(\psi_1(\boldsymbol{x}) - \psi_2(\boldsymbol{y}))^\top \boldsymbol{\Lambda_x} (\psi_1(\boldsymbol{x}) - \psi_2(\boldsymbol{y}))$ |
| AdaSSL | $-\lambda(\psi_1(\boldsymbol{x}, \hat{\boldsymbol{r}}) - \psi_2(\boldsymbol{y}))^\top (\psi_1(\boldsymbol{x}, \hat{\boldsymbol{r}}) - \psi_2(\boldsymbol{y}))$ |

**Complex $p(\mathbf{c}^+ \mid \mathbf{c})$, formally stated.**

$$\boldsymbol{\kappa} \sim \mathcal{N}(0, \Sigma), \quad \mathrm{c}_i \mid \boldsymbol{\kappa} \sim \mathcal{N}(\mu(\boldsymbol{\kappa})_i, \sigma(\boldsymbol{\kappa})_i^2), \tag{16}$$

$$\iota_i \mid \boldsymbol{\kappa} \sim \mathrm{Bern}(\pi(\boldsymbol{\kappa})_i), \quad \mathrm{c}_i^+ \mid \iota_i, \mathrm{c}_i, \boldsymbol{\kappa} \sim \begin{cases} \delta(\mathrm{c}_i^+ = \mathrm{c}_i), & \iota_i = 0 \\ \mathcal{N}(\mu(\boldsymbol{\kappa})_i, \sigma(\boldsymbol{\kappa})_i^2), & \iota_i = 1 \end{cases}. \tag{17}$$

**Data.** We set $n_c = n_s = 5$ and sample $\Sigma \sim \mathcal{W}^{-1}(n_c + 2, \mathbf{I})$. For anisotropic noise, we sample $\sigma(\mathbf{c})_i^2 \sim \mathrm{InvGamma}(2, 1)$. For heteroscedastic noise, we set $\sigma(\mathbf{c})^2 = \mathrm{softplus}(\mathbf{W}_\sigma \mathbf{c} + \mathrm{softplus}^{-1}(1))$. For complex $p(\mathbf{c}^+ \mid \mathbf{c})$, we use $\mu(\boldsymbol{\kappa}) = \mathbf{W}_\mu^\top \boldsymbol{\kappa} + \boldsymbol{b}$, $\sigma(\boldsymbol{\kappa})^2 = \mathrm{softplus}(\mathbf{W}_\sigma \boldsymbol{\kappa} + \mathrm{softplus}^{-1}(1))$, and $\pi_i(\boldsymbol{\kappa}) = \mathrm{Sigmoid}\left(\frac{\kappa_i}{\Sigma_{ii}} - 1\right)$. We sample each element of $\mathbf{W}_\mu, \mathbf{W}_\sigma$, and $\boldsymbol{b}$ from $\mathcal{N}(0, 1)$. We parameterize $g_{\mathrm{MLP}}$ as a three-layer MLP with `LeakyReLU` activation (negative slope 0.2) with the same number of units in all layers. We ensure invertibility by using $L^2$-normalized weight matrices that has the lowest condition number among 25 000 uniformly sampled candidates. We use $\mathbf{x}^{++} = g_{\mathrm{MLP}}([\mathbf{c}^+, \mathbf{s}^{++}])$ where $\mathbf{c}^+$ is the same content factor as in $\mathbf{x}^+$ and $\mathbf{s}^{++} \sim \mathcal{N}(0, \mathbf{I})$.

**Architecture.** For the encoder $f$, we use an MLP with four hidden layers of dimensionality $10n$ where $n = n_c + n_s$ is the input dimension. For models that apply $L^2$ normalization to the outputs, we set the output dimensionality to $n + 1$ to accommodate for the missing degree of freedom; otherwise we set it to $n$. For H-InfoNCE$_{\mathrm{Affine}}$, we use an affine layer followed by `softplus` activation to predict $\Lambda_\mathbf{x}$. For H-InfoNCE$_{\mathrm{MLP}}$, we use an MLP with three hidden layers of size $10n$ followed by `softplus` activation to predict $\Lambda_\mathbf{x}$ and an MLP of the same size to predict $\phi_1(\mathbf{x})$ in Table 2. For AdaSSL, we set $d_r = 5$. We use MLPs with two hidden layers of dimension 64 to parameterize $q_\phi$, $p_\theta$, and $m$ and use a linear $t$ for AdaSSL-V. All MLPs except the encoder use a `BatchNorm` layer followed by `LeakyReLU` with the default negative slope ($0.01$) after each hidden layer.

**Hyperparameters.** We use the `AdamW` optimizer (Loshchilov & Hutter, 2019) with learning rate $5 \times 10^{-4}$ and weight decay $10^{-4}$ on the parameters except biases. We use a batch size of 2048. For the experiments on complex $p(\mathbf{c}^+ \mid \mathbf{c})$, we apply the loss symmetrically similar to Chen et al. (2020a) because the sampling process of $\mathbf{c}$ and $\mathbf{c}^+$ is symmetric. We train the models for 200 000 steps and observe convergence. For AdaSSL-V, we linearly warmup $\beta$ from 0 to 0.5 for 1000 steps to prevent early KL instabilities. We keep $\beta = 1$ fixed throughout training for AdaSSL-S. For the unimodal $p(\mathbf{c}^+ \mid \mathbf{c})$ experiments, we set $\tau = \mathbb{E}[\sigma_i^2(\mathbf{c})] = 1$ except when the variance is fixed to 0, in which case we set $\tau = 0.1$. For the complex $p(\mathbf{c}^+ \mid \mathbf{c})$ experiments, we set $\tau = 0.1$.

**Evaluation.** We perform evaluation by training a linear regressor on top of the frozen representations on 100 000 unseen data samples and evaluate it on another 100 000 samples.

**Hardware.** Each trial of this experiment required approximately 15-20 hours to run, using eight CPU cores, 4 GB of system memory, and an MIG-partitioned slice of an NVIDIA H100 GPU providing roughly a quarter of the GPU's compute capacity and 20 GB of GPU memory.

### G.3 CRL experiments in §C

**Data.** 3DIdent contains $250\,000$ training images in $\mathcal{D}_{\text{train}}$ and $25\,000$ test images in $\mathcal{D}_{\text{test}}$, which we use for CRL experiments. We sample latent pairs $(\mathbf{z}, \mathbf{z}^+)$ following

$$\mathbf{z} \sim p(\mathbf{z})\,,\ \tilde{\mathbf{z}} \sim p(\tilde{\mathbf{z}})\,,\ \iota_i \sim \text{Bern}(0.2)\,,\ z_i^+ = z_i \text{ if } \iota_i = 0 \text{ else } z_i^+ = \tilde{z}_i \text{ for } i \in [d_z]\,. \quad (18)$$

Since 3DIdent is a finite dataset, after obtaining a latent pair, we find their nearest neighbor in the training set with FAISS (Douze et al., 2024) and use the correspondingly rendered observations as inputs following the original authors (Zimmermann et al., 2021). AdaSSL does not use an additional view in this experiment.

**Data augmentations.** For standard pairs, we use the same set of strong augmentations used for CelebA. For natural pairs, we do not perform augmentations. We resize the images to $128 \times 128$ resolution.

**Architecture.** We use a ResNet-18 encoder followed by a two layer MLP projector with hidden size of 128 and output size of 16, and `ReLU` activation without `BatchNorm` as $f$. For AdaSSL, we set $d_r = 16$. We use MLPs with two hidden layers of dimension 128 to parameterize $q_\phi$, $p_\theta$, and $m$. These MLPs use a `BatchNorm` layer followed by `ReLU` activation after each hidden layer. As discussed in §C, we ablate the parameterization of $t$ for AdaSSL-V; the MLP parameterization has a hidden layer of dimensionality 128 with `BatchNorm` followed by `ReLU` activations. The VAE-based methods use a ResNet-18 decoder that mirror the encoder.

**Hyperparameters.** We use the `AdamW` optimizer with learning rate $10^{-4}$, weight decay $10^{-5}$ on non-bias parameters, and a batch size of 256. For contrastive learning, we calculate the loss symmetrically following standard practice (Chen et al., 2020a). We train all models for $150\,000$ steps and observe convergence on $\mathcal{D}_{\text{train}}$. All SSL methods use a normalized embedding space, use $\tau = 0.05$, and do not learn $\lambda$ in this experiment. For AdaSSL-V, we perform linear warmup of $\beta$ from 0 to 0.5 for $10\,000$ steps to prevent early KL instabilities. For AdaSSL-S, we fix $\beta = 0.5$. For AdaGVAE, we search within the authors' recommended set of $\beta$'s, [1, 2, 4, 8, 16], but find $\beta = 100$ to give the best disentanglement.

**Evaluation.** We perform evaluation on $\mathcal{D}_{\text{test}}$ with the frozen embeddings and ground-truth latent factors with linear regression and the DCI disentanglement score. We normalize the embeddings for the SSL based models such that they align with the training objective, similar to Zimmermann et al. (2021). We use the posterior mean as the embeddings for VAE-based models and do not normalize them. For the DCI disentanglement score, we use the weights of Lasso regressors as the relative importance matrix.

**Hardware.** Each trial of this experiment required approximately 15-20 hours to run, using eight CPU cores, 32 GB of system memory, and an MIG-partitioned slice of an NVIDIA H100 GPU providing roughly three-eighths of the GPU's compute capacity and 40 GB of GPU memory.

### G.4 Natural image experiments in §3.3

**Data.** We split the CelebA dataset into $\mathcal{D}_{\text{train}}$, $\mathcal{D}_{\text{val}}$, and $\mathcal{D}_{\text{test}}$ following an 8-1-1 ratio; this gives us $161\,908$ training images, $20\,346$ images in the validation set and $20\,345$ images in the test set. To create a natural distribution shift, we sample celebrity identity such that the people in $\mathcal{D}_{\text{train}}$ does not appear in $\mathcal{D}_{\text{val}} \cup \mathcal{D}_{\text{test}}$. This gives us 8142 celebrities in $\mathcal{D}_{\text{train}}$ and 2035 celebrities in $\mathcal{D}_{\text{val}} \cup \mathcal{D}_{\text{test}}$. To construct a structured positive pair, we randomly sample two images of the same person. This results in $1\,850\,918$ possible positive pairs. Data pairs examples are visualized in Fig. 4 and the distribution of the number of differed attributes between pairs are shown in Fig. 5, confirming that attributes differ sparsely between positive pairs. During training, we augment the sampled pair using data augmentations and obtain $\mathbf{x}$ and $\mathbf{x}^+$. We use another augmented view of $\mathbf{x}^+$ as $\mathbf{x}^{++}$. This is helpful because our goal is not to learn the low-level style factors, but instead the semantic content factors that differ structurally between $\mathbf{x}^+$ and $\mathbf{x}$. The standard pairing process still use augmented versions of the same image as positive pairs.

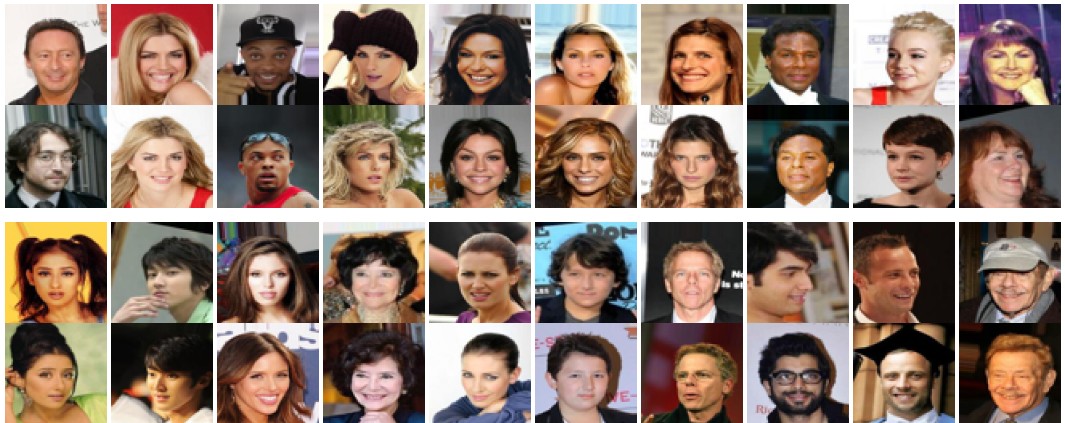

Figure 4: Visualization of images paired by identity from the CelebA dataset.

**Data augmentations.** We investigate the effect of both strong and weak augmentations. For strong augmentations, we apply the standard set of augmentations used in SSL studies (Chen et al., 2020a; Grill et al., 2020). We use `RandomHorizontalFlip` with 0.5 probability, then `RandomResizedCrop` with crops of size within $[8\%, 100\%]$ of the original image and aspect ratio within $[0.75, 1.33]$, which are then resized to $64 \times 64$. Next, with probability $0.8$, we randomly apply `ColorJitter` where the brightness, contrast, saturation and hue of the image are shifted by a uniformly random offset. We use parameters $0.4, 0.4, 0.2, 0.1$, respectively. Finally, we apply `RandomGrayScale` with probability $0.2$, `GaussianBlur` with probability $0.5$, and `Solarization` with probability $0.2$. For weak augmentations, we only apply `RandomHorizontalFlip` with probability of $0.5$

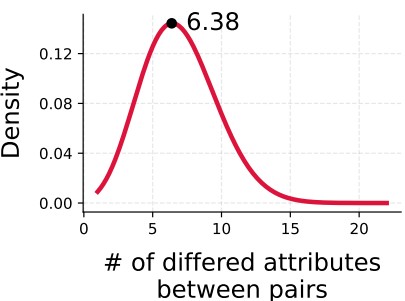

Figure 5: Distribution of the number of differed attributes between pairs of images of the same identity.

and `RandomResizedCrop` with crops of size within $[80\%, 100\%]$ of the original image and aspect ratio within $[0.9, 1.1]$. Notice that this cropping operation is significantly weaker than the one used for strong augmentations.

**Architecture.** We use a ResNet-18 encoder (He et al., 2016) followed by a two layer MLP projector with hidden size of 1024 and output size of 128, and `ReLU` activation without `BatchNorm` as $f$ similar to Chen et al. (2020a). For AdaSSL, we set $d_r = 20$. We use MLPs with one hidden layer of dimension 1024 to parameterize $q_\phi$, $p_\theta$, and $m$. We use an MLP with one hidden layer of dimension 512 to parameterize $t$ for AdaSSL-V; this MLP does not have a bias term in the output layer, similar to the predictor in BYOL (Grill et al., 2020). These MLPs use a `BatchNorm` layer followed by `ReLU` activation after each hidden layer.

**Hyperparameters.** We use the `AdamW` optimizer with learning rate $2 \times 10^{-4}$ and weight decay $10^{-4}$ on the parameters except biases. We use a batch size of 512. For contrastive learning, we calculate the loss symmetrically following standard practice (Chen et al., 2020a). We train the models for $80\,000$ steps and observe convergence on $\mathcal{D}_{\text{val}}$. All models use a normalized embedding space and use $\tau = 0.1$. For AdaSSL-V, we perform linear warmup of $\beta$ from 0 to 0.1 for $10\,000$ steps to prevent early KL instabilities. For AdaSSL-S, we fix $\beta = 0.5$.

**Evaluation.** Following standard practice, we train a linear classifier with the `BinaryCrossEntropy` loss for each attribute on top of the frozen representations and embeddings on $\mathcal{D}_{\text{train}}$ until convergence and evaluate it on $\mathcal{D}_{\text{test}}$. We use the $F_1$ score of the minority class as the evaluation metric because the attributes are highly imbalanced. To do that, we compute the $F_1$ score for each attribute then report the mean score over attributes.

**Hardware.**    Each trial of this experiment required approximately 15-20 hours to run, using 12 CPU cores, 24 GB of system memory, and an NVIDIA L40S GPU with 48 GB of GPU memory.

### G.5    Video experiments in §3.3

**Data.**    We construct a custom dataset similar to Moving-MNIST (Srivastava et al., 2015; Drozdov et al., 2024), where nine-frame videos are generated stochastically on the fly from sample images in MNIST. For a given image, we first create a black $64 \times 64$ canvas. Afterwards, we resize the original $28 \times 28$ image to $16 \times 16$ and place it on the canvas after uniformly sampling its initial center coordinates from $[8, 16]$. In frames 1-3, the digit moves from this center based on a velocity in the horizontal direction, denoted by $v_{x,1:3}$, and in the vertical direction, denoted by $v_{y,1:3}$. We sample these initial velocities uniformly from $[0, v_0]$ where $v_0 = 3$. Then, with an equal probability, we sample one direction and change its velocity by adding a Gaussian noise proportional to the initial velocity (i.e., heteroscedastic):

$$\iota \sim \mathrm{Bern}(0.5)\,, \quad \begin{cases} v_{x,4:9} \sim \mathcal{N}\left(v_{x,1:3}, \frac{2}{3}v_{x,1:3}\right), v_{y,4:9} = v_{y,1:3}\,, & \iota = 0 \\ v_{y,4:9} \sim \mathcal{N}\left(v_{y,1:3}, \frac{2}{3}v_{y,1:3}\right), v_{x,4:9} = v_{x,1:3}\,, & \iota = 1 \end{cases}. \tag{19}$$

This makes the new velocity in frame 4-9 within $(-v_0, 3v_0)$ with high probability. Generated video samples are shown in Figure 6. We refer to this as *Setting A*.

In *Setting B*, we let $\iota$ depend on the digit input. Concretely, we use equally spaced bins between 0.1 and 0.9 for the ten digits:

$$\iota_k \sim \mathrm{Bern}(p_k)\,, \quad \text{where} \quad p_k = 0.1 + k \cdot \frac{0.9 - 0.1}{10 - 1}\,, \quad k = 0, \ldots, 9\,. \tag{20}$$

This means the distribution of the direction of acceleration varies for different digits.

We partition each sampled video into three-frame segments and use them as $\mathbf{x}$, $\mathbf{x}^+$, and $\mathbf{x}^{++}$ (§G.1). The model predicts $f(\mathbf{x}^+)$ from $f(\mathbf{x})$ (and optionally $f(\mathbf{x}^{++})$ by AdaSSL and BYOL+Future). The goal is to capture both the digit class and the velocity in the three-frame video representations. We partition the $60\,000$ MNIST images into $50\,000$ training images and $10\,000$ validation images and use each set for generating training and validation videos on the fly. Note that we always sample the velocities online, and the model observes different videos in every epoch.

**Architecture.**    The encoder $f$ consists of a 3D convolutional encoder, followed by an MLP projector. The 3D convolutional encoder consists of five convolutional layers with $[32, 64, 128, 128, 256]$ channels with `BatchNorm` and `ReLU` activations after each layer. The first two and the last layer have spatial-only kernels of dimensions $[1, 3, 3]$ and the third and fourth layers have temporal convolutions with kernels of dimensions $[3, 1, 1]$. The encoder outputs are average-pooled on the spatial dimensions and then flattened across the temporal dimension resulting in a 768-dimensional representation. The representations are passed to an MLP projector with two hidden layers of size 1024, each followed by `BatchNorm` and `ReLU` activations. The output embeddings have a dimensionality of 128, and are batch-normalized. The projector is followed by an MLP predictor $h$ with two hidden layers of dimensionality 1024 with `BatchNorm` and `ReLU` activations after each hidden layer. The predictor output does not use `BatchNorm` or `ReLU`. For AdaSSL-V, we use a two-dimensional $\mathbf{r}$, which is concatenated to $f(\mathbf{x})$ as the predictor input. We use MLPs with one hidden layer of dimensionality 1024 to parameterize $q_\phi$, $p_\theta$, and $m$. These MLPs use a `BatchNorm` layer followed by `ReLU` activation after each hidden layer. For BYOL+Future, we concatenate the projector embeddings $f(\mathbf{x})$ and $f(\mathbf{x}^{++})$ and use it as the predictor input. BYOL+GT predicts $f(\mathbf{x}^+)$ from $f(\mathbf{x})$ and $r^\star$, the ground-truth difference between the velocities of $\mathbf{x}$ and $\mathbf{x}^+$. We experiment with concatenating $r^\star$ directly with $f(\mathbf{x})$ or passing it through a learnable linear embedding before concatenation, and find that using an embedding layer slightly improves performance.

**Hyperparameters.**    For all methods, we train the model for $75\,000$ steps with the `AdamW` optimizer using a batch size of 128. We use an initial learning rate of $10^{-4}$ and decay it following a cosine schedule, following Grill et al. (2020). We use a constant weight decay of $10^{-4}$. For the EMA momentum, we use a constant decay rate of 0.996. In BYOL+GT, we learn an affine projection to create an embedding for $\mathbf{r}^\star$ of dimensionality 32. For all AdaSSL models, we use a constant regularization coefficient $\beta$, and in our default setting, $d_r = 2$ and $\beta = 0.001$.

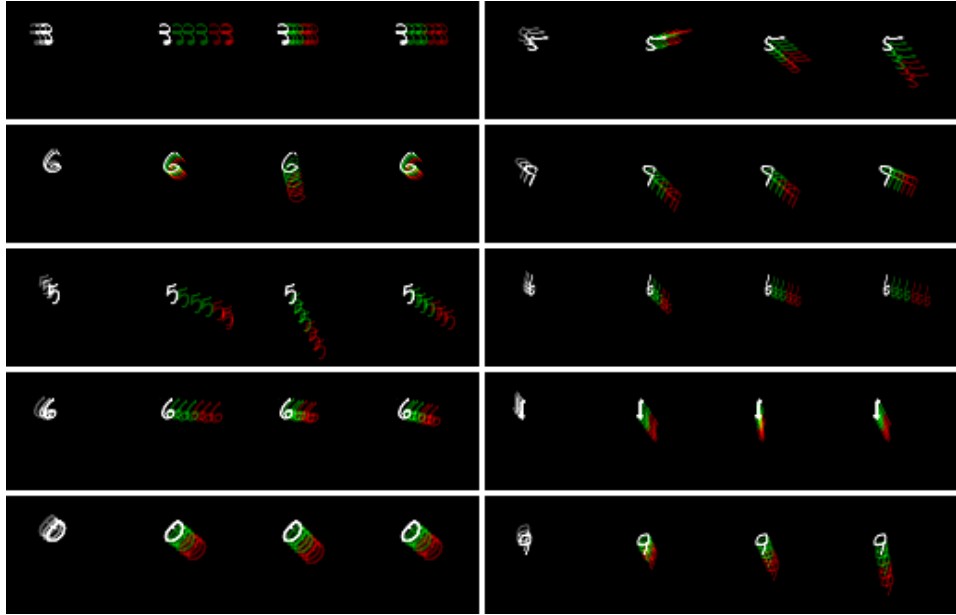

Figure 6: Random samples (nine-frame video sequences) from the stochastic Moving-MNIST dataset. For each example, the first three frames (context) are shown on the left. Then, three different future trajectories of the next six frames (targets) are randomly sampled according to Eq. 19 and visualized to the right of the initial three-frame segment. The third frame is overlaid on all canvases for reference. The motion uncertainty arises from random velocity changes along spatial directions.

Table 6: Performance of linear probes trained on frozen representations and embeddings on stochastic Moving-MNIST. Evaluation is performed on the online branch of BYOL.

| Model | SETTING A | | | | SETTING B | | | |
| | Representations | | Embeddings | | Representations | | Embeddings | |
| | Acc. [%] | Velocity [$R^2$] | Acc. [%] | Velocity [$R^2$] | Acc. [%] | Velocity [$R^2$] | Acc. [%] | Velocity [$R^2$] |
|---|---|---|---|---|---|---|---|---|
| BYOL | $90.42_{\pm 0.94}$ | $0.8753_{\pm 0.0044}$ | $87.09_{\pm 2.41}$ | $0.1079_{\pm 0.0061}$ | $91.00_{\pm 1.07}$ | $0.8810_{\pm 0.0057}$ | $88.61_{\pm 1.79}$ | $0.1486_{\pm 0.0303}$ |
| BYOL+Future | $88.31_{\pm 1.14}$ | $0.9005_{\pm 0.0063}$ | $78.68_{\pm 0.55}$ | $0.5890_{\pm 0.0242}$ | $88.33_{\pm 1.09}$ | $0.8996_{\pm 0.0059}$ | $78.99_{\pm 0.45}$ | $0.6041_{\pm 0.0186}$ |
| BYOL+GT | $93.09_{\pm 0.24}$ | $0.8814_{\pm 0.0078}$ | $88.95_{\pm 0.56}$ | $-0.0038_{\pm 0.0060}$ | $93.55_{\pm 0.50}$ | $0.8884_{\pm 0.0062}$ | $87.99_{\pm 0.36}$ | $-0.0028_{\pm 0.0045}$ |
| AdaSSL-V$_{\beta=0}$ | $\mathbf{94.18}_{\pm 0.51}$ | $0.8951_{\pm 0.0066}$ | $\underline{90.54}_{\pm 0.54}$ | $0.2867_{\pm 0.0184}$ | $\underline{94.17}_{\pm 0.19}$ | $0.8961_{\pm 0.0028}$ | $\underline{90.34}_{\pm 0.66}$ | $0.2875_{\pm 0.0219}$ |
| AdaSSL-V | $\underline{93.83}_{\pm 0.22}$ | $\mathbf{0.9168}_{\pm 0.0015}$ | $\mathbf{91.28}_{\pm 0.43}$ | $\underline{0.8695}_{\pm 0.0185}$ | $\mathbf{94.31}_{\pm 0.48}$ | $\mathbf{0.9188}_{\pm 0.0006}$ | $\mathbf{92.32}_{\pm 0.73}$ | $\underline{0.8594}_{\pm 0.0035}$ |
| AdaSSL-S | $91.89_{\pm 0.74}$ | $\underline{0.9121}_{\pm 0.0028}$ | $86.00_{\pm 0.33}$ | $\mathbf{0.8901}_{\pm 0.0247}$ | $91.95_{\pm 0.53}$ | $\underline{0.9121}_{\pm 0.0032}$ | $85.53_{\pm 1.90}$ | $\mathbf{0.8750}_{\pm 0.0121}$ |

**Evaluation.** To perform evaluation, we train linear probes with `CrossEntropy` (for digit classification) and `MSE` (for velocity regression) losses on top of the frozen video representations and embeddings of the online branch on $\mathcal{D}_{\text{train}}$ until convergence. We then report the digit prediction accuracy and velocity decoding $R^2$ scores on a fixed video test set generated from the 10 000 test images of MNIST.

**Hardware.** Each trial of this experiment required approximately 6-8 hours to run, using six CPU cores, 32 GB of system memory, and an NVIDIA H100 GPU with 80 GB of GPU memory.

# H  Additional results

## H.1  Density

To understand why AdaSSL outperforms baselines in Table 2, we visualize the aggregated marginal distribution of $\mathbf{z}^+$ implied by the learned predictor, $\mathbb{E}_{\mathbf{z}}[p_{\text{model}}(\mathbf{z}^+ \mid \mathbf{z})]$, using Monte-Carlo estimates from true pairs $p(\mathbf{z}, \mathbf{z}^+)$ (Fig. 7). For InfoNCE, we first encode the input $\mathbf{x} = g(\mathbf{z})$ and then learn a projection from the embedding space to the ground-truth latent space by training a linear regressor from $f(g(\mathbf{z}))$ to $\mathbf{z}^+$. For H-InfoNCE, we pass $f(g(\mathbf{z}))$ through the predictor and project

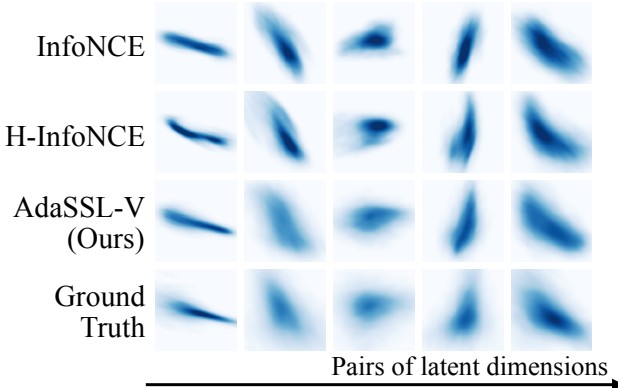

Pairs of latent dimensions

Figure 7: Aggregated marginal distributions $\mathbb{E}_{\mathbf{z}}[p_{\text{model}}(\mathbf{z}^+ \mid \mathbf{z})]$ across latent dimension pairs. InfoNCE produces collapsed densities and H-InfoNCE partially recovers variability, while AdaSSL-V aligns closely with the ground truth. The improvement is most evident in columns two and three, where AdaSSL-V captures both spread and orientation while baselines do not.

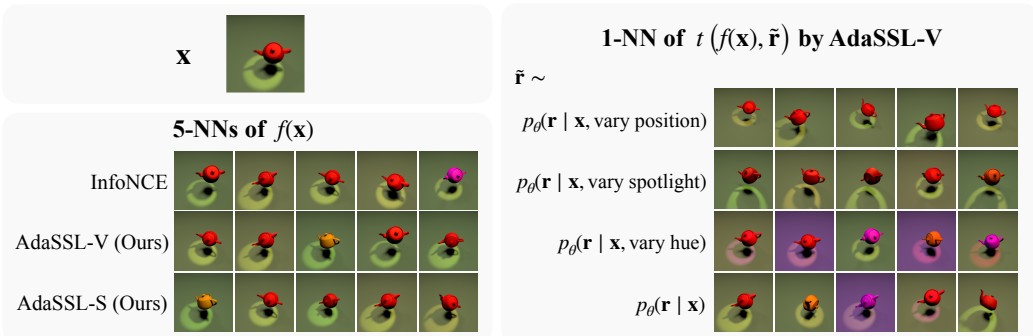

Figure 8: Image retrieval results on 3DIdent. Top left: query image. Bottom left: five nearest neighbors on the embeddings. Right: controllable retrieval by AdaSSL-V.

the predicted representations. For AdaSSL-V, we sample from the learned prior $\tilde{\mathbf{r}} \sim p_\theta(\mathbf{r} \mid \mathbf{x})$ and use $\tilde{\mathbf{r}}$ to edit the embeddings with $t(f(\mathbf{x}), \tilde{\mathbf{r}})$ and project the edited embeddings. InfoNCE embeddings produce overly concentrated densities, indicating their inability to accurately capture complex conditional uncertainties. H-InfoNCE partially corrects this, while AdaSSL best fits the ground-truth distribution, suggesting that its improvement arises from more accurate modeling of the conditional uncertainty.

## H.2 Retrieval

In Fig. 8 (left), we perform standard retrieval to accompany our analysis in §C. We retrieve the five nearest neighbors of the query image in the embedding space. We observe that both AdaSSL and the baselines are able to retrieve visually similar images. There are still some wrong retrievals in color and spotlight, and rotation is especially hard to learn for all methods.

## H.3 Stochastic Moving-MNIST

We provide full evaluation results on stochastic Moving-MNIST in Table 6. These results further demonstrate AdaSSL's effectiveness in achieving strong performance in both digit recognition and velocity decoding. Our results and ablations in the main text in Fig. 1 uses Setting A because we do not find significant difference between the results.

