# OpenReview forum: "Self-Supervised Learning from Structural Invariance"
_NeurIPS.cc/2025/Workshop/UniReps — UniReps2025_

### Official Review · Reviewer_jsVW · 2025-09-15
**A novel SSL method with strong theory and early empirical promise, needing larger-scale validation.**

**Confidence:** 3

**Review:**

### Summary

This paper presents AdaSSL - an extension of Joint-Embedding Self-supervised Learning (SSL) approaches that learns to model the uncertainty of matching positive pairs in the latent space. The approach is motivated by the fact that in real-world datasets, there may exist multiple valid matches between samples. The approach has two variants and is applicable to contrastive and non-contrastive SSL approaches. The authors demonstrate the efficacy of AdaSSL through experiments on numerical, image, and video data.

### Strengths

1. The work is a novel addition to the active area of research on learning the right invariances in SSL.
2. The authors provide a strong theoretical justification for AdaSSL.
3. Compared to previous works on this topic, AdaSSL does not assume the availability of the parametrization of the “style” data (e.g. augmentations), but learns it implicitly.
4. The paper inspects a variational and sparse variant of AdaSSL, confirming the usefulness of the variational approach.
5. The initial experiment results on numerical and high-dimensional data are encouraging. For example, AdaSSL shows improvement over pure InfoNCE loss even under weak augmentations.

### Weaknesses

1. I suggest adding a figure depicting the approach, as it would significantly clarify the method.
2. The $\mathbf{r}$ variable is crucial in section 2.2, but is defined very loosely, several lines after being used for the first time. Introducing it before Line 76 would greatly improve the readability of the paper.
3. The experiments are relatively low-scale, and I would expect evaluations on a larger dataset, at least ImageNet-100, in the standard Joint-Embedding SSL setup.
4. While AdaSSL is applicable to both contrastive and non-contrastive frameworks, the rationale for using a contrastive backbone on images and a non-contrastive one (BYOL) on videos is not clearly explained.

### Justification of the verdict

While the preliminary experiments are very low-scale and the paper being trimmed to fit the 4 pages somewhat limits the readability, the work discusses a timely topic in SSL, describes an intriguing idea and the preliminary results are encouraging.
I believe the paper will be of interest to the UniReps workshop community.

**Score:**

4

**Topic Fit:**

3

---

### Official Review · Reviewer_5Ht7 · 2025-09-15
**Review of "Self-Supervised Learning from Structural Invariance"**

**Confidence:** 3

**Review:**

The paper proposes Adaptive Self-Supervised Learning (AdaSSL), a framework to address the one-to-many mapping problem in self-supervised learning from naturally paired data. Standard SSL methods assume simple conditional distributions between positive pairs, which breaks down when it is multimodal or heteroscedastic (e.g., video futures, image–caption variability). AdaSSL introduces an auxiliary latent variable r to capture uncertainty in the conditional mapping. There are two variants presented, AdaSSL-V, a variational formulation, and AdaSSL-S, a deterministic formulation with a sparsity penalty. Theoretical results show that heteroscedasticity is unavoidable in SSL embeddings under general conditions. Empirically, AdaSSL outperforms InfoNCE, AnInfoNCE, and BYOL variants on synthetic data, CelebA facial attributes, and Moving-MNIST videos, particularly in settings with complex conditional structure.

Strengths:
- The paper identifies an important gap in SSL, the inability of current methods to model complex conditional distributions (heteroscedastic, multimodal), especially in naturally paired data. This framing is clear.
- Novelty: Introducing a latent variable r inferred from pairs (rather than assumed available, as in JEPAs) is to the best of my knowledge an as of yet unexplored direction. Additionally, the authors provide proof that heteroscedasticity is inevitable in SSL embeddings.
- The authors evaluate their method on numerical data, images, and videos, which demonstrates the method’s versatility across domains.
- The work is structured well and the language is clear and appropriate.

Weaknesses:
- The results section could be clearer. For example, the paper claims that (b) and (c) evaluate robustness under distribution shifts, but only (c) is a true OOD setup. It is not entirely clear how the authors themselves view case (b).
- The text refers to InfoNCE and AnInfoNCE results in Table 1, but these rows are not actually present, which creates confusion for the reader.
- The paper reads more like a full conference submission than a workshop extended abstract, which may make it harder for readers to quickly grasp the main ideas.

Overall, this is a very strong and well-motivated paper that makes a clear contribution, and with the minor clarifications and tweaks mentioned above, it could spark valuable discussion at the workshop.

**Score:**

4

**Topic Fit:**

3

---

### Official Review · Reviewer_Z5gB · 2025-09-15
**Paper proposes to enhance contrastive learning by learning positive pair uncertainty even when under heteroscedastic noise**

**Confidence:** 5

**Review:**

The paper proposes a "probabilistic" alternative to contrastive learning to improve the ability of these representation learning approaches to learn the underlying distribution p(z^+|z) including when taking non trivial structures (i.e.,heteroscedastic noise).

Strengths:
- The empirical results show the effect of the proposed method on relevant tasks.
- Topic is relevant to the community
- Paper is quite well written and clear.

Weaknesses:
- For a full paper more ablations on the training and test distribution hyper parameters (in the numerical setup) should be included as well as multiple random seeds.
- World modeling experiments would benefit from more baselines


Suggestions:
- typo line 87
- what do $\phi$ represent in equation 4?
- I think [1] and [2] should be included as references as they aim to explicitly model p(z^+|z) via a latent variable model.

[1] Sinha, Samarth, and Adji Bousso Dieng. "Consistency regularization for variational auto-encoders." Advances in Neural Information Processing Systems 34 (2021): 12943-12954.

[2] Bizeul, Alice, Bernhard Schölkopf, and Carl Allen. "A Probabilistic Model behind Self-Supervised Learning." Transactions on Machine Learning Research.

**Score:**

4

**Topic Fit:**

2

---

### Official Review · Reviewer_TCfa · 2025-09-16
**Summary**

**Confidence:** 3

**Review:**

**Summary**

The authors propose to learn structured invariance for naturally paired data by learning a latent variable that encodes uncertainty between pairs. The proposed loss is approximately a lower-bound on the mutual information between different views, and the method outperforms other baselines in numerical and high-dimensional natural statistics data.

**Strength**

- The connection to IT and the lower bound observation are a good read.
- The paper is well-written and easy to follow.

**Weakness**

- It would be great to know what are the extra costs involved in optimizing for the latent variable r when we vary different axes of scaling like model width etc.
- It might be better to explain the experimental details for the high-dimensional data in the main text. Right now it’s not very clear to me how do you parameterize everything in your loss.

**Score:**

4

**Topic Fit:**

2

---

### Official Review · Reviewer_9QHy · 2025-09-16
**Review of `Self-Supervised Learning from Structural Invariance`**

**Confidence:** 4

**Review:**

This work aim to address a limitation of current SSL approaches, which assume a unimodal mapping between paired samples, whereas many real world data sources have paired samples in a "one-to-many" fashion. The authors introduce AdaSSL, a method that augments SSL methods with a learned latent variable that captures residual uncertainty in the positive pair relationship.

**Originality**
High - the limitation they seek to address is well-known: the assumption that positive pairs should collapse to a single embedding. The proposed framework is a general approach can be applied to both conditional and predictive SSL, allowing for flexible modeling of conditional uncertainty in positive pairs.

**Quality**
High - this work is theoretically motivated, with the authors providing a mutual information decomposition approach and derivation of a tractable variational bound. Both variants of AdaSSL (V/S) are presented with clear justification, and experiments cover multiple data settings to establish the generality of this method.

**Significance**
High - this work provides a unified way to handle the aforementioned limitation and is compatible with existing contrastive/predictive methods. Ablations clearly demonstrate the power of this method compared to baselines.

**Weaknesses**
* This work does not include experimentation on large-scale, real world datasets (e.g., ImageNet).
* I would've like to see further analysis of $r$ to see what it actually encodes.
* Similar works include multi-positive InfoNCE and JEPAs with action-conditioning - a comparsion to this methods may be warranted.
* Additional compute/training time analysis for AdaSSL v. vanilla SSL.

Overall, excellent job!

**Score:**

5

**Topic Fit:**

3

---

### Decision · Program_Chairs · 2025-09-21

Accept (oral)